# SkelHCC: A Hyperbolic CLIP-Driven Cache Adaptation Framework for Skeleton-based One-Shot Action Recognition

Yanan Liu [* 1]   Anqi Zhu [* 2]   Jingmin Zhu [2]   Jun Liu [3]   Hossein Rahmani [3]   Mohammed Bennamoun [4]
Farid Boussaid [4]   Dan Xu [1]   Qiuhong Ke [2]

## Abstract

Skeleton-based action recognition aims to understand human behaviors from body joint sequences and is especially challenging in the one-shot setting, where only a single labeled exemplar is available for each novel action. A key challenge is learning representations that capture the hierarchical and compositional structure of human motion while aligning effectively with high-level action semantics under extreme data scarcity. Existing approaches, largely based on Euclidean embeddings and low-level motion cues, struggle to model the tree-like organization of skeleton data, limiting cross-modal alignment and generalization to unseen action categories. We propose SkelHCC, a unified skeleton hyperbolic CLIP-driven cache adaptation framework for one-shot skeleton-based action recognition. SkelHCC introduces an Explicitly Hierarchical Hyperbolic CLIP (EH-HCLIP) module that embeds skeleton sequences and action language into a shared hyperbolic space. By leveraging the negative curvature and exponential volume growth of hyperbolic geometry, EH-HCLIP naturally encodes the joint–part–body hierarchy of human anatomy and yields structurally consistent cross-modal representations. To support efficient one-shot adaptation, SkelHCC further integrates a training-free LLM-guided Multi-granularity Voting Cache (LMV-Cache) for context-aware inference. Experiments on NTU RGB+D 60, NTU RGB+D 120, and PKU-MMD II demonstrate that SkelHCC consistently outperforms state-of-the-art methods. Code will be available at github.com/SkelHCC.

---

[*]Equal contribution  [1]Yunnan University, Kunming, China [2]Monash University [3]Lancaster University [4]University of Western Australia. Correspondence to: Dan Xu <danxu@ynu.edu.cn>, Qiuhong Ke  <qiuhong.ke@monash.edu>.

*Proceedings of the $43^{rd}$ International Conference on Machine Learning*, Seoul, South Korea. PMLR 306, 2026. Copyright 2026 by the author(s).

## 1. Introduction

With the advancement of affordable depth sensors and powerful pose estimation models, skeleton-based action recognition has emerged as an active research area in recent years (Shi et al., 2019; Cheng et al., 2020; Zhou et al., 2024), owing to its robustness, computational efficiency, and invariance to environmental variations. Existing methods mainly focus on exploring fully-supervised skeleton-based action recognition requiring large-scale, high-cost annotations. However, in many real-world domains such as healthcare or rehabilitation (Liu et al., 2023), collecting and labeling motion data from patients or elderly subjects is costly and time-consuming. These challenges have motivated research into **O**ne-shot **S**keleton-based **A**ction **R**ecognition (**OSAR**) (Liu et al., 2020a), where the model is trained on seen classes with sufficient samples and must recognize novel, unseen actions from only a single exemplar per class. This paradigm aims to achieve strong generalization under extreme data scarcity, a setting that pushes beyond the capabilities of conventional supervised skeleton recognition.

Despite recent progress, OSAR remains difficult due to two fundamental challenges: (1) Representation alignment challenge: Human actions are inherently hierarchical and compositional: coordinated joint motions form body parts, which in turn interact to produce full-body behaviors. This hierarchical organization induces structured, tree-like dependencies in skeleton data that are difficult to encode and align with high-level action semantics. Consequently, existing methods (Memmesheimer et al., 2021; 2022; Wang & Koniusz, 2022b) that rely mainly on low-level motion cues from single exemplars struggle to align structured skeleton representations with language-based action semantics, hindering generalization to unseen categories. (2) Adaptation challenge: Even with semantically aligned representations, one-shot generalization requires leveraging the exemplar efficiently at test time. Models must identify which body regions are most relevant for discriminating between actions and perform context-aware similarity matching rather than relying on global features alone.

To address these challenges, we propose **SkelHCC** — a unified **Skel**eton **H**yperbolic **C**LIP-driven **C**ache adaptation

framework that couples hierarchical representation alignment and adaptive inference within a single hyperbolic manifold.

Specifically, to tackle the representation alignment challenge, SkelHCC incorporates an **E**xplicitly **H**ierarchical **H**yperbolic **CLIP (EH-HCLIP)** module that embeds skeleton and language modalities into a shared hyperbolic space. Recent studies (Franco et al., 2023; Li et al., 2025; Tu et al., 2025) have shown that skeleton data exhibits strong compatibility with hyperbolic geometry, motivating its use for structured motion modeling. To further substantiate this choice, we analyze the $\delta$-hyperbolicity (Ganea et al., 2018a) of skeleton graphs, confirming their non-Euclidean, tree-like characteristics (see Appendix I). Unlike conventional CLIP operating in flat Euclidean geometry, EH-HCLIP leverages the negative curvature and exponential volume growth of hyperbolic space to encode the tree-like hierarchy of human anatomy (joint $\rightarrow$ part $\rightarrow$ full-body). This design explicitly aligns the bio-anatomical structure of skeletons with the semantic hierarchy of action language, producing structurally consistent representations essential for one-shot recognition.

To address the adaptation challenge, SkelHCC integrates a **LLM-guided Multi-granularity Voting Cache (LMV-Cache)**, a memory-based inference module that reuses the learned hyperbolic space for test-time adaptation. LMV-Cache stores support exemplars as cache entries and employs Large Language Model (LLM)-derived semantic masks to emphasize discriminative joints and parts based on the textual description of each action. Leveraging EH-HCLIP–learned weights, LMV-Cache fuses multi-granularity similarities through weighted voting to achieve context-aware, training-free inference on unseen actions.

We test our SkelHCC framework on three benchmark datasets NTU RGB+D 60 (Shahroudy et al., 2016), NTU RGB+D 120 (Liu et al., 2020a), and PKU-MMD II (Liu et al., 2020b). Extensive experiments demonstrate the effectiveness of our approach. Notably, SkelHCC outperforms the SOTA methods (Yan et al., 2024) by margins of 6.7% and 9.0% under the 20 and 100 base class settings on the NTU RGB+D 120 dataset.

We summarize our key contributions as follows:

- We propose SkelHCC, a cache-driven hyperbolic CLIP framework that unifies hierarchical representation alignment and adaptive inference within a consistent geometric–linguistic space for OSAR.

- SkelHCC integrates two new complementary components: EH-HCLIP for hierarchical skeleton–language alignment in hyperbolic space and LMV-Cache for LLM-guided adaptation, enabling multi-granularity similarity fusion and context-aware inference on un-

seen actions.

- Our SkelHCC achieves state-of-the-art performance on three challenging datasets, outperforming the existing SOTA methods by a significant margin.

## 2. Related Work

### 2.1. Skeleton-based Action Recognition

**Fully-supervised skeleton-based action recognition** primarily focuses on network designing (Cui & Hayama, 2024)—such as Recurrent Neural Networks (RNNs) (Du et al., 2015), Convolutional Networks (CNNs) (Tas & Koniusz, 2018), and Graph Convolutional Networks (GCNs) (Cheng et al., 2020; Peng et al., 2024; Liu et al., 2025)—to capture the spatial topology and temporal motion dependencies in skeleton data. Many state-of-the-art methods (Zhou et al., 2024; Yan et al., 2018; Shi et al., 2019; Liu et al., 2020c) aim to strike a balance by introducing a "threshold" mechanism that selectively retains or discards spatial interactions based on their semantic relevance. These methods provide effective backbones for skeletal feature extraction in one-shot learning. Recently, Li et al. (Li et al., 2025) proposed a hyperbolic linear attention tailored to skeleton data, validating the superiority of hyperbolic space over Euclidean space for modeling the hierarchy of skeleton-based actions.

**One-shot Skeleton-based Action Recognition** is garnering significant attention from the computer vision community. Some methods build a metric space from seen data to reframe classification as nearest neighbor search. Wang et al. (Wang & Koniusz, 2022b) used uncertainty DTW to measure the temporal distance of skeleton sequences. Memmesheimer et al. (Memmesheimer et al., 2021) proposed a signal level deep metric learning (SL-DML) approach for one-shot action recognition. Another approach is to design strategies to enhance the discriminability of action representations. Memmesheimer et al. (Memmesheimer et al., 2022) proposed Skeleton-DML, which converts skeletons into discriminative pseudo image representations. Liu et al. (Liu et al., 2020a) pioneered the Action Part Semantic Relevance Perception (APSR) framework for OSAR on the NTU RGB+D 120 dataset, using networks to calculate attention scores for each human component and weighting them to focus on discriminative regions.

The latest CrossGLG (Yan et al., 2024) believes that the advanced human semantics generated by LLM can guide a attention generator to learn the correct joint distribution from the skeleton feature. However, these advanced knowledges are not directly applied to test samples with a semantic bias. In other words, LLMs just provide priors for training data, but the generator operates on test features, preventing distribution adaptation from training to testing. By con-

trast, our LMV-Cache directly incorporates LLM-derived body-joint/part masks to similarity assessment between test samples and cache values, focusing on critical body regions to effectively mitigate semantic bias. Furthermore, most methods (Memmesheimer et al., 2022; Wang & Koniusz, 2022b) treat skeleton representations as a whole, the hierarchical mechanism in our SkelHCC highlights multi-grained discriminative features to improve one-shot generalization.

## 2.2. Vision-language Representation Learning

Linguistic descriptions from LLMs provide complementary high-level semantics and context, effectively disambiguating visual inputs (Lee, 2025; Wei et al., 2022). Representation learning methods like CLIP (Radford et al., 2021), as pioneered by visual-language pretraining, providing richer supervisory signals and exhibit strong generalization in zero-shot tasks (Zhu et al., 2026a; Chen et al., 2024; Zhu et al., 2024). Representatively, (Zhu et al., 2026b) targets training-free adaptation for skeleton action recognition, it focuses on zero-shot, test-driven self-adaptation with an online pseudo-label cache, while SkelHCC leverages language for multi-granularity support–query matching and hyperbolic skeleton–text retrieval.

Some methods (Zhang et al., 2022; Gao et al., 2024; Zhou et al., 2022) have demonstrated CLIP's potential for few-shot learning by leveraging fine-tuning or cache model (Khandelwal et al., 2020; Grave et al., 2017; Zhang et al., 2022) techniques. Hyperbolic space offers inherent advantages over Euclidean space in modeling hierarchical structures (Khrulkov et al., 2020), with applications expanding from natural language processing (NLP) (Dhingra et al., 2018; Tifrea et al., 2019) to pixel-based vision tasks such as image segmentation (Atigh et al., 2022) and anomaly detection (Li et al., 2024a). Inspired by CLIP, Desai et al. (Desai et al., 2023) built an image-text hyperbolic representation to better capture the underlying hierarchy. However, these methods are designed for images or videos rather than skeleton data. Considering that tree-like human skeleton, we introduce the EH-HCLIP module for one-shot learning, constructing a skeleton-text hyperbolic-aligned space with tailored designs to adapt the skeleton modality.

## 3. Method

Here, we will introduce the proposed SkelHCC, a novel one-shot skeleton action recognition framework based on skeleton-text hyperbolic CLIP space and LLM-guided cache model. We begin by briefly reviewing the necessary preliminaries in Section 3.1. Section 3.2 presents an overview of the architecture, with detailed descriptions of each component provided in Section 3.3 and Section 3.4.

## 3.1. Preliminaries

**Lorentz Hyperbolic Model** serves as a computationally favorable representation of hyperbolic geometry (Krioukov et al., 2010), with its core idea being the definition of vectors on a hyperboloid embedded in a high-dimensional *Minkowski* spacetime (Peng et al., 2022; Desai et al., 2023). In other words, the $d$-dimensional hyperbolic space is realized by taking the upper sheet of a two-sheeted hyperboloid in $(d+1)$-dimensional Minkowski space $\mathbb{R}^{d+1}$. Specifically, the vector $\mathbf{x} \in \mathbb{R}^{d+1}$ can be formulated as a spatio-temporal form $\{\mathbf{x}_{spa}, x_{tem}\}$, where $\mathbf{x}_{spa} \in \mathbb{R}^d$ and $x_{tem} \in \mathbb{R}$. Intuitively, the symmetry axis of the hyperboloid corresponds to the temporal dimension, while the remaining axes represent spatial dimensions. Therefore, a key *Lorentz inner product* can be defined as:

$$\langle \mathbf{x}, \mathbf{y} \rangle_{\mathbb{L}} = -x_{tem}y_{tem} + \langle \mathbf{x}_{spa}, \mathbf{y}_{spa} \rangle, \quad (1)$$

where $\langle \cdot, \cdot \rangle_{\mathbb{L}}$ denotes the *Lorentzian inner product* and $\langle \cdot, \cdot \rangle$ denotes the *Euclidean inner product*. The $d$-dimensional hyperbolic space with a curvature $c$ is defined as:

$$\mathbb{L}^d = \left\{ \mathbf{x} = \{\mathbf{x}_{spa}, x_{tem}\} \in \mathbb{R}^{d+1} : \langle \mathbf{x}, \mathbf{x} \rangle_{\mathbb{L}} = -1/c \right\},$$
$$x_{tem} = \sqrt{\frac{1}{c} + \|\mathbf{x}_{spa}\|^2}, \quad (2)$$

where $\|\cdot\|^2$ denote the the square of the *Euclidean* norm. All vectors in the hyperbolic space $\mathbb{L}^d$ need to satisfy the above constraints. Based on this, *Lorentzian* distance between two points can be formulated as:

$$d_{\mathbb{L},c}(\mathbf{x}, \mathbf{y}) = \sqrt{\frac{1}{c}} \operatorname{arcosh}\left(-c\langle \mathbf{x}, \mathbf{y} \rangle_{\mathbb{L}}\right). \quad (3)$$

The *Lorentzian* distance $d_{\mathbb{L},c}(\mathbf{x}, \mathbf{y})$ with curvature $c$ is used to measure the shortest distance in hyperbolic space.

***Base* and *Novel* Class.** One-shot action recognition is based on two non-overlapping datasets (Liu et al., 2020a; Zhu et al., 2023a; Yan et al., 2024; Memmesheimer et al., 2021; 2022; Wang & Koniusz, 2022b): a labeled *base* class dataset $\mathcal{D}_{base}$ and a *novel* class dataset $\mathcal{D}_{novel}$. $\mathcal{D}_{novel}$ comprises $N_{novel}$ classes, one exemplar from each class and its corresponding label form the *support* set, while the remaining samples form the *query* set.

## 3.2. Overall Architecture

As illustrated in Fig 1, the proposed cache adaptation framework for OSAR, SkelHCC, comprises two core components: **1)** A novel Explicit Hierarchical Hyperbolic CLIP (EH-HCLIP) $f_{EH-HCLIP}(\cdot)$, which is a contrastive learning approach that integrates a skeleton hierarchical bio-anatomical prior, achieving robust skeleton–text semantic alignment in hyperbolic space. **2)** A carefully designed

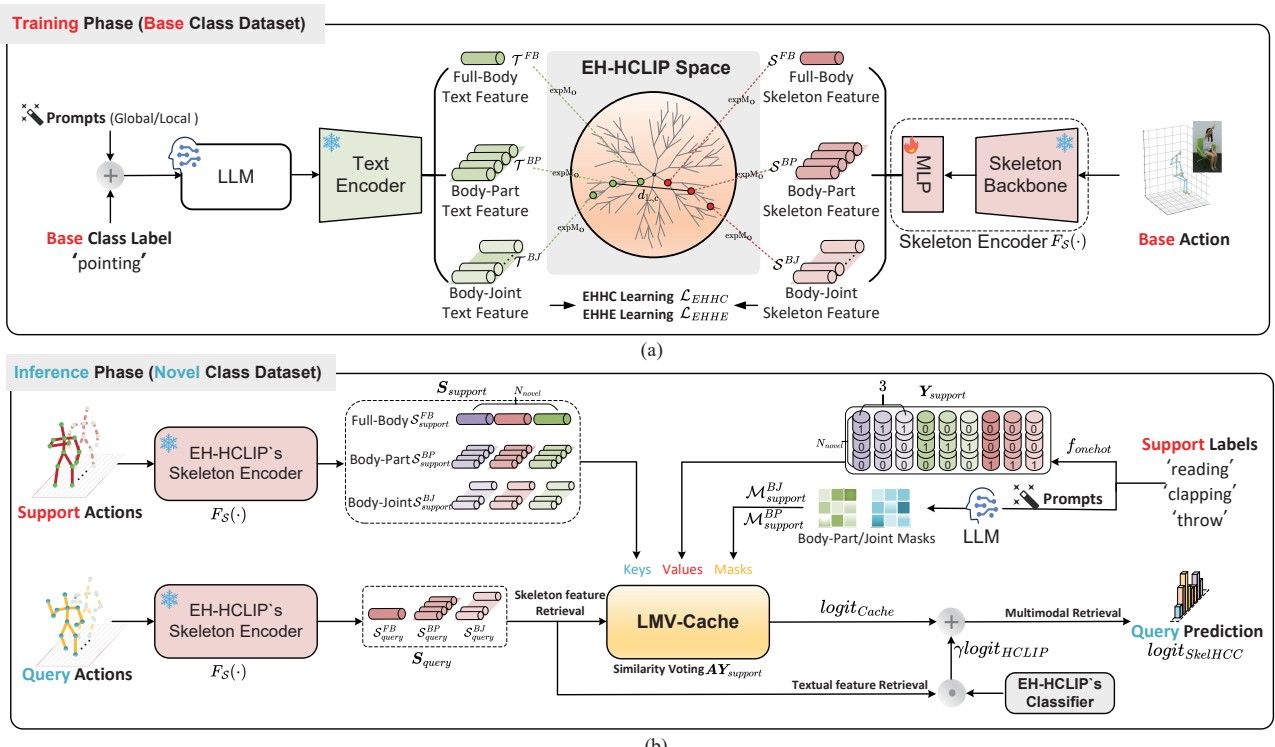

*Figure 1.* Overall Architecture of our proposed SkelHCC. (a) The training pipeline of the SkelHCC. We use the frozen CLIP's Text Encoder (Radford et al., 2021) and a frozen skeleton backbone (Chen et al., 2021) pretrained on the *base* class dataset. (b) The inference pipeline of the SkelHCC. The calculation of $logit_{HCLIP}$ is shown in Eq. 10. More details of prompts are provided in the *Appendix*.

LLM-guided Multi-granularity Voting Cache module (LMV-Cache) $f_{LMV-Cache}(\cdot)$, which is a training-free cache model stores features of support actions and their labels as a key-value database, incorporating specialized mechanisms tailored for skeleton data to enable effective similarity-based retrieval. The pipeline of SkelHCC is as follows:

During training, the EH-HCLIP module and the *base* class dataset are utilized to learn a skeleton-text aligned hyperbolic CLIP space. EH-HCLIP employs a frozen CLIP's text encoder (Radford et al., 2021) and a pre-trained skeleton backbone (Chen et al., 2021; Cheng et al., 2020), requiring only the training of an MLP interface for the skeleton modality, thus introducing negligible training cost.

During inference, classification is transformed into a multimodal retrieval. On the *novel* class dataset, the features extracted by EH-HCLIP's skeleton encoder from query samples are used to achieve 1) a skeleton-skeleton similarity-based retrieval from LMV-Cache to generate a cache logit $logit_{Cache}$ and 2) a skeleton-text similarity-based retrieval via EH-HCLIP to generate a HCLIP logit $logit_{HCLIP}$.

Finally, SkelHCC combines the $logit_{Cache}$ and the $logit_{HCLIP}$ that provides more reliable linguistic supervision via a residual connection (He et al., 2016), enabling a robust OSAR prediction. Below, we describe the details.

### 3.3. Explicitly Hierarchical Hyperbolic CLIP

We strive to customize a hyperbolic CLIP space to provide additional linguistic supervision for OSAR. Human skeleton exhibits an inherent hierarchical structure, in which the semantics of certain actions are localized within specific body regions (Liu et al., 2020a). Therefore, we leverage these hierarchical relationships to improve alignment. Guided by bio-anatomical priors, we define three granularity levels: 1) *Body Joint* (BJ), 2) *Body Part* (BP), and 3) *Full Body* (FB). Based on the above three granularities, we introduce an explicitly hierarchical hyperbolic contrastive learning method to achieve better skeleton-text alignment.

**Projecting Embeddings onto the Hyperboloid.** Given a skeleton feature set $\{(\mathcal{S}_k, \mathcal{Y}_k)\}_{k=1}^K$ with $K$ action samples, where $\mathcal{S}$ and $\mathcal{Y}$ denote the skeleton feature generated by the skeleton encoder $F_\mathcal{S}(\cdot)$ and the corresponding label, respectively. $F_\mathcal{S}(\cdot)$ incorporates a frozen backbone (Chen et al., 2021; Cheng et al., 2020; Zhou et al., 2024), pretrained on the *base* class dataset, followed by an MLP head. First, we performed a skeleton partitioning based on prior knowledge of human kinematics (Liu et al., 2020a) (see details in *Appendix C*), yielding multi-grained skeleton feature $\boldsymbol{S}_k = \{\mathcal{S}_k^{FB} \in \mathbb{R}^C, \mathcal{S}_k^{BJ} \in \mathbb{R}^{C \times V}, S_k^{BP} \in \mathbb{R}^{C \times V_p}\}$, where $\mathcal{S}_k^{FB}, \mathcal{S}_k^{BJ}$ and $\mathcal{S}_k^{BP}$ denote the full-body, body-joint

and body-part skeleton features. $V$ and $V_p$ are the number of body joints and parts. $C$ is the number of channels. Next, we design different prompts to generate both global and local textual action descriptions by leveraging LLMs (e.g. GPT-4), yielding multi-grained text feature $\boldsymbol{T}_k = \{\mathcal{T}_k^{FB} \in \mathbb{R}^C, \mathcal{T}_k^{BJ} \in \mathbb{R}^{C \times V}, \mathcal{T}_k^{BP} \in \mathbb{R}^{C \times V_p}\}$ via the CLIP text encoder (Radford et al., 2021), where $\mathcal{T}_k^{FB}, \mathcal{T}_k^{BJ}$ and $\mathcal{T}_k^{BP}$ denote the full-body, body-joint and body-part text features (See in the *Appendix* F). Finally, the skeleton hyperbolic embedding set $\tilde{\boldsymbol{S}}_k = \{\tilde{\mathcal{S}}_k^{FB}, \tilde{\mathcal{S}}_k^{BJ}, \tilde{\mathcal{S}}_k^{BP}\} = \mathrm{expM_O}(\boldsymbol{S}_k)$ and text hyperbolic embedding set $\tilde{\boldsymbol{T}}_k = \{\tilde{\mathcal{T}}_k^{FB}, \tilde{\mathcal{T}}_k^{BJ}, \tilde{\mathcal{T}}_k^{BP}\} = \mathrm{expM_O}(\boldsymbol{T}_k)$ are generated by the exponential map $\mathrm{expM_O}(\cdot)$, as shown in *Appendix* B, where $\mathbf{O}$ denotes the origin of the hyperboloid.

**Explicitly Hierarchical Hyperbolic Contrastive (EHHC) learning.** Given a skeleton, we construct positive and negative samples to minimize distance between paired skeleton-text pairs and maximize distance between unpaired ones. The positive sample is its paired text, while the negative samples are all other texts in the batch. To achieve contrastive learning loss, guided by (Radford et al., 2021; Desai et al., 2023), we employ the negative *Lorentzian* distance $d_{\mathbb{L},c}(\cdot)$ with curvature c in Eq.3 to measure the similarity between skeleton and text embeddings. The resulting logits are then transformed into a probability distribution via a softmax function. The formulation is described as follows:

$$\mathcal{L}_{HCL}(\tilde{\mathcal{S}}, \tilde{\mathcal{T}}) = -\sum_{i=1}^{B} \log \frac{\exp\left(-d_{\mathbb{L},c}(\tilde{\mathcal{S}}_i, \tilde{\mathcal{T}}_i)/\tau\right)}{\sum_{j=1}^{B} \exp\left(-d_{\mathbb{L},c}(\tilde{\mathcal{S}}_i, \tilde{\mathcal{T}}_j)/\tau\right)},$$
(4)

where $\mathcal{L}_{HCL}$ denotes the hyperbolic contrastive loss. $B$ is batch size. $\tilde{\mathcal{S}}$ and $\tilde{\mathcal{T}}$ are the skeleton and text hyperbolic embedding. To achieve bidirectional alignment, the $\mathcal{L}_{HCL}(\tilde{\mathcal{T}}, \tilde{\mathcal{S}})$ is formulated by constructing the negative samples from all non-matching skeletons within the same batch. Next, we introduce a Explicitly Hierarchical Hyperbolic Contrastive (EHHC) loss $\mathcal{L}_{EHHC}$ as

$$\mathcal{L}_{EHHC} = \sum_{i=1}^{|\tilde{\boldsymbol{S}}|} \frac{\alpha_i}{2}\Big(\mathcal{L}_{HCL}(\tilde{\boldsymbol{S}}^{(i)}, \tilde{\boldsymbol{T}}^{(i)}) + \mathcal{L}_{HCL}(\tilde{\boldsymbol{T}}^{(i)}, \tilde{\boldsymbol{S}}^{(i)})\Big),$$
(5)

where $|\tilde{\boldsymbol{S}}|$ denotes the number of elements in the skeleton hyperbolic embedding set $\tilde{\boldsymbol{S}}$. $\tilde{\boldsymbol{S}}^{(i)}$ and $\tilde{\boldsymbol{T}}^{(i)}$ represent the skeleton and text hyperbolic embedding, which achieves multi-level supervision, explicitly highlighting features from local details to global context. $\alpha_i$ represents learnable weights to dynamically adjust the impact of three granularities. In addition, followed by (Desai et al., 2023; Ganea et al., 2018b), we also introduce a Hyperbolic Entailment Learning (HEL) loss $\mathcal{L}_{HEL}$, which leverages the geometric constraints of entailment cones to enforce the partial order (Vendrov et al., 2015) between the paired skeleton and text. More details

about $\mathcal{L}_{HEL}$ are described in the *Appendix* D. Based on this, we also construct an Explicit Hierarchical Hyperbolic Entailment (EHHE) loss $\mathcal{L}_{EHHE}$ as

$$\mathcal{L}_{EHHE} = \sum_{i=1}^{|\tilde{\boldsymbol{S}}|} \alpha_i \mathcal{L}_{HEL}(\tilde{\boldsymbol{S}}^{(i)}, \tilde{\boldsymbol{T}}^{(i)}).$$
(6)

Ultimately, hyperbolic semantic alignment between skeleton and text is trained by EH-HCLIP loss $\mathcal{L}_{EH-HCLIP}$ as

$$\mathcal{L}_{EH-HCLIP} = \mathcal{L}_{EHHC} + \lambda \mathcal{L}_{EHHE} + \mathcal{L}_{CE}, \quad (7)$$

where $\mathcal{L}_{CE}$ denotes the cross-entropy loss between the output logits and the ground-truth labels to boost generalization by penalizing misclassification, and $\lambda$ is a hyperparameter to balance this objective.

### 3.4. LLM-guided Multi-granularity Voting Cache

So far, we have trained an aligned EH-HCLIP space. However, it did not leverage the exemplar efficiently at test time. Existing cache models (Zhang et al., 2022) are usually designed for images and do not fully consider the characteristics of skeleton data. Therefore, we introduce a training-free LMV-Cache module tailored for skeleton data, which acts as an adapter for EH-HCLIP to boost the OSAR performance.

**One-shot Cache Construction.** Given the *novel* class dataset $D_{novel}$ with $K_{novel}$ action samples from $N_{novel}$ classes. We aim at constructing a key-value cache model incorporating the one-shot knowledge of the $N_{novel}$ classes, which can be viewed as a feature adapter.

For *support* action samples, we use the EH-HCLIP's skeleton encoder $F_{\mathcal{S}}(\cdot)$ to extract a high-dimensional graph representation and still perform a hierarchical partitioning. The multi-granularity *support* skeleton features $\boldsymbol{S}_{support}$ and corresponding label vectors $\boldsymbol{Y}_{support}$ are described as:

$$\boldsymbol{S}_{support} = \{\mathcal{S}_{support}^{FB}, \mathcal{S}_{support}^{BJ}, \mathcal{S}_{support}^{BP}\},$$
$$\boldsymbol{Y}_{support} = \{f_{onehot}(\mathcal{Y}_{support})\}_{i=1}^{3},$$
(8)

where $\mathcal{S}_{support}^{FB} \in \mathbb{R}^{N_{novel} \times C}$, $\mathcal{S}_{support}^{BJ} \in \mathbb{R}^{N_{novel} \times C \times V}$, $S_{support}^{BP} \in \mathbb{R}^{N_{novel} \times C \times V_p}$ denote the *support* skeleton features from three granularity levels, $V$ is the number of body joints, $V_p$ is the number of body parts and $C$ is the number of channels, while its corresponding label $\mathcal{Y}_{support}$ is converted into a one-hot vector $\boldsymbol{Y}_{support} \in \mathbb{R}^{3N_{novel} \times N_{novel}}$. In our proposed LMV-Cache, $\boldsymbol{S}_{support}$ are treated as the cache keys, while their corresponding one-hot labels $\boldsymbol{Y}_{support}$ are treated as the cache values.

It should be noted that $\boldsymbol{S}_{support}$ is a skeleton feature set derived from a single exemplar at three distinct granularity levels. This way allows for the more comprehensive *novel* class knowledge to be preserved within this cache model, thereby providing a more robust similarity-based retrieval.

**Body-joint mask prompt**

Given the action [**Taking a selfie**], provide the representativeness (degree of contribution) of each body part when recognizing the action, in percentages (0.0–1.0). The sum should equal 1.0. Each body part's importance should be provided in the following order: base of spine, middle of spine, ...

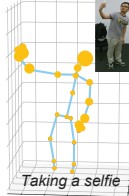

*Taking a selfie*

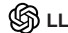 **LLM Output: Body-joint mask value**

base of spine: **0.01**, middle of spine: **0.02**, neck: **0.05**, head: **0.18**, left shoulder: **0.04**, left elbow: **0.06**, left wrist: **0.07**, left hand: **0.08**, right shoulder: **0.04**, right elbow: **0.08**, right wrist: **0.09**, right hand: **0.1**, left hip: **0.01**, left knee: **0.0**, left ankle: **0.0**, left foot: **0.0**, right hip: **0.01**, right knee: **0.0**, right ankle: **0.0**, right foot: **0.0**, spine: **0.03**, tip of left hand: **0.03**, left thumb: **0.02**, tip of right hand: **0.04**, right thumb: **0.03**

*Figure 2.* The diagram of prompting LLMs (GPT 4) to obtaining body-joint mask value, combined with visualization of a weight on the skeleton. Body-part masks and more details can be found in the *Appendix*.

**LLM-guided Similarity Retrieval.** Based on this cache model, query samples can be classified through a straightforward similarity retrieval, without requiring complex model fine-tuning or additional parameters. To utilize more valuable information from one-shot samples, we introduce linguistic priors by prompting the LLMs, as shown in Fig 2. These priors take the form of the body-joint importance masks $\mathcal{M}^{BJ}_{support} \in \mathbb{R}^{N_{novel} \times V}$ and body-part importance masks $\mathcal{M}^{BP}_{support} \in \mathbb{R}^{N_{novel} \times V_p}$, highlighting the critical body areas, thereby optimizing the similarity retrieval.

Specifically, given a query sample, generating skeleton feature set $\boldsymbol{S}_{query} = \{\mathcal{S}^{FB}_{query} \in \mathbb{R}^{1 \times C}, \mathcal{S}^{BJ}_{query} \in \mathbb{R}^{1 \times C \times V}, S^{BP}_{query} \in \mathbb{R}^{1 \times C \times V_p}\}$ can be regarded as a cache query. The affinities between cache query and keys are calculated as follows:

$$Sim^{FB} = \alpha_1 \varphi(\mathcal{S}^{FB}_{query}, \mathcal{S}^{FB}_{support}),$$
$$Sim^{BJ} = \alpha_2 \frac{1}{V} \sum_{i=1}^{V} \left[ \phi(\mathcal{S}^{BJ}_{query}, \mathcal{S}^{BJ}_{support}) \odot \mathcal{M}^{BJ}_{support} \right]_i,$$
$$Sim^{BP} = \alpha_3 \frac{1}{V_p} \sum_{i=1}^{V_p} \left[ \phi(\mathcal{S}^{BP}_{query}, \mathcal{S}^{BP}_{support}) \odot \mathcal{M}^{BP}_{support} \right]_i,$$
$$\boldsymbol{A} = \exp(-\beta(1 - Concat(Sim^{FB}, Sim^{BJ}, Sim^{BP})),$$
$$(9)$$

where $\boldsymbol{A} \in \mathbb{R}^{1 \times 3N_{novel}}$ denotes the affinity score. $\varphi(\cdot)$ denotes dot product. $\phi(\cdot)$ and $\odot$ denote the Hadamard product for the calculation of the similarity $Sim$. $V$ and $V_p$ denote the number of body joints and parts. $\alpha_1, \alpha_2, \alpha_3$ represent the weights of three granularities derived from EH-HCLIP. $\beta$ denotes a modulating hyper-parameter. LMV-Cache generates an accurate cache logit $logit_{Cache} = \boldsymbol{A}\boldsymbol{Y}_{support} \in \mathbb{R}^{1 \times N_{novel}}$ through a matrix multiplication with the cache values $\boldsymbol{Y}_{support} \in \mathbb{R}^{3N_{novel} \times N_{novel}}$, which enables a multi-granularity voting over the three similarity candidates, reformulating the hard prediction reliant on a single exemplar

in one-shot dataset as a robust soft prediction.

In addition, the EH-HCLIP plays as a classifier during inference to enable the similarity assessment between query skeleton features and support textual features to generate the HCLIP logit $logit_{HCLIP}$ as:

$$logit_{HCLIP} = -\sum_{i=1}^{|\tilde{S}|} \alpha_i d_{\mathbb{L},c}(\tilde{\boldsymbol{S}}^{(i)}_{query}, \tilde{\boldsymbol{T}}^{(i)}_{support}), \quad (10)$$

where $d_{\mathbb{L},c}(\cdot)$ denotes the *Lorentzian* distance with curvature c. $\tilde{\boldsymbol{S}}^{(i)}_{query}$ and $\tilde{\boldsymbol{T}}^{(i)}_{support}$ denote the query hyperbolic skeleton embedding and support hyperbolic text embedding, respectively. The final prediction logit $logit_{SkelHCC}$ of query samples can be formulated as:

$$logit_{SkelHCC} = logit_{Cache} + \gamma logit_{HCLIP}, \quad (11)$$

where $\gamma$ denotes the residual weight. $logit_{SkelHCC}$ includes two items, the former $logit_{Cache}$ represents similarity-based retrieval between skeleton features on LVM-Cache, the latter $logit_{HCLIP}$ leverages linguistic priors to perform similarity retrieval between skeleton features and textual features in aligned EH-HCLIP space. $\gamma$ controls the trade-off between them. In fact, the aligned EH-HCLIP can also be regarded as a special textual cache model.

## 4. Experiments

### 4.1. Datasets

We evaluated the proposed methods on three large skeleton-based action datasets: **1) NTU-RGB+D 60** (NTU60) (Shahroudy et al., 2016) is a large-scale human action dataset containing 56,880 samples, which contains 60 action labels and 25 major joints, including daily, interactive and health-related action. One-shot evaluation settings include 10 novel classes. **2) NTU-RGB+D 120** (NTU120) (Liu et al., 2020a) is the most authoritative skeleton-based action dataset, especially in one-shot evaluation, which extends NTU60, totaling 114,480 samples from 106 subjects across 120 classes. This dataset contains a total of 20 novel classes under the one-shot evaluation settings. **3) PKU-MMD II** (P-MMD) (Liu et al., 2020b) is another large-scale action dataset with 41 classes containing more than 6,900 sequences, with the viewpoint variation presenting the primary challenge for robust action analysis.

### 4.2. Implementation Details

Our method is implemented by *Python* and *Pytorch* on a single *RTX 2080 Ti* GPU. Our training is to align the EH-HCLIP space. The skeleton encoder consists of a frozen backbone pretrained on the *base* class dataset followed by (Chen et al., 2021; Zhou et al., 2024; Lee et al., 2023;

*Table 1.* Comparison of the performance with state-of-the-arts on NTU RGB+D 120 dataset for one-shot skeleton-based action recognition. **Bold** highlights the best performance. Underline indicates the second-highest performance. Params (Adapt.) indicate the number of parameters optimized during one-shot adaptation, while Total Params denote the full model size. By freezing the pretrained backbone and training only 0.5M adaptation parameters, our method enables fast one-shot adaptation.

| Base Classes | 20 | 40 | 60 | 80 | 100 | Backbone Updated | Params (Adapt.) | Total Params |
|---|---|---|---|---|---|---|---|---|
| APSR (Liu et al., 2020a) | 29.1 | 34.8 | 39.2 | 42.8 | 45.3 | ✓ | – | – |
| uDTW (Wang & Koniusz, 2022b) | 32.2 | 39.0 | 41.2 | 45.3 | 49.0 | ✓ | – | – |
| SL-DML (Memmesheimer et al., 2021) | 36.7 | 42.4 | 49.0 | 46.4 | 50.9 | ✓ | 11.8 | 11.8 |
| Skeleton-DML (Memmesheimer et al., 2022) | 28.6 | 37.5 | 48.6 | 48.0 | 54.2 | ✓ | 11.8 | 11.8 |
| JEANIE (Wang & Koniusz, 2022a) | 38.5 | 44.1 | 50.3 | 51.2 | 57.0 | ✓ | – | – |
| SMAM (Li et al., 2023) | 35.8 | 46.2 | 51.7 | 52.2 | 56.4 | ✓ | – | – |
| ALCA-GCN (Zhu et al., 2023a) | 38.7 | 46.6 | 51.0 | 53.7 | 57.6 | ✓ | – | – |
| MotionBERT (Zhu et al., 2023b) | 35.5 | 54.3 | 56.5 | 52.8 | 61.0 | ✓ | 60.3 | 60.3 |
| Trans4SOAR (Peng et al., 2023) | – | – | – | – | 57.1 | ✓ | 43.8 | 43.8 |
| STA-MLN (Li et al., 2024b) | 42.5 | 48.8 | 53.1 | 54.3 | 59.9 | ✓ | – | – |
| MC-Scale (Yang et al., 2024) | 44.1 | 55.3 | 60.3 | 64.2 | 68.7 | ✓ | 15.1 | 15.1 |
| SkeletonX (Zhang et al., 2025) | 48.2 | 54.9 | 61.6 | 65.6 | 69.1 | ✓ | 1.6 | **1.6** |
| GAP (Xiang et al., 2023) | 35.1 | 54.8 | 50.9 | 53.3 | 59.9 | ✓ | 1.6 | **1.6** |
| CrossGLG (Yan et al., 2024) | 45.3 | 56.8 | 62.1 | 61.6 | 62.6 | ✓ | 1.7 | 1.7 |
| **SkelHCC(Ours)** | **52.0** | **60.9** | **67.4** | **67.9** | **71.6** | ✗ | **0.5** | 1.8 |

Cheng et al., 2020) and a trainable MLP, while the frozen CLIP (Radford et al., 2021) is adopted as the text encoder. The details of text description and prompts are described in the *Appendix*. Each action sample is resized to 70 frames. We employed the *Adam* optimizer. The learning rate is set to 0.0001. The batch size is set to 20. The curvature parameter is set to 0.1. The hyperparameter $\beta$ and $\lambda$ are set to 1.0 and 0.1. The parameters $\alpha_1, \alpha_2, \alpha_3$ are initially set to 10.5.

**Evaluation Protocol** followed the (Liu et al., 2020a; Yan et al., 2024) without any additional optimization strategies, using *base* class dataset for training and reserving the *novel* class dataset for evaluation—with only one fixed exemplar per *novel* class. For NTU120, we use the 100/20 *base*/*novel* split from (Liu et al., 2020a); for NTU60 and P-MMD, we assign 10 novel classes with one fixed exemplar followed by (Yan et al., 2024; Liu et al., 2020b).

### 4.3. Comparison with the State-of-the-arts

Table 1 compares the recognition performance with SOTA methods under different numbers of *base* classes on the NTU120 dataset. Our proposed SkelHCC outperforms these SOTA methods significantly in all evaluation metrics. We attribute this improvement to our cache adaptation framework, which provides effective multi-modal supervision and substantially enhances the model's generalization. Specifically, our approach significantly outperforms ALCA-GCN (Zhu et al., 2023a) (+13.3% under the 20-class setting and +14.0% under the 100-class setting). We also surpass the accuracy of the prominent motion understanding method, Motion-Bert (Zhu et al., 2023b), by 10.6% under the 100-class setting, while achieving a significant parameters reduction.

Additionally, we have compared with two representative skeleton-text multimodal approaches, GAP (Xiang et al., 2023) and CrossGLG (Yan et al., 2024). Our SkelHCC significantly outperforms GAP (Xiang et al., 2023) – the text-guided method designed for the fully supervised setting, evaluated under a one-shot scenario – by 11.7% in the

*Table 2.* Comparison of the performance with state-of-the-arts on NTU RGB+D 60 dataset for one-shot skeleton-based action recognition. **Bold** highlights the best performance. Underline indicates the second-highest performance.

| Method | NTU60 | P-MMD |
|---|---|---|
| uDTW (Wang & Koniusz, 2022b) | 72.4 | – |
| CATFormer (Long, 2023) | 73.2 | 31.8 |
| FEAT (Ye et al., 2020) | 74.3 | 32.3 |
| SMAM (Li et al., 2023) | 73.6 | - |
| FR-Head (Zhou et al., 2023) | 78.1 | 34.3 |
| CrossGLG (Yan et al., 2024) | 75.6 | - |
| MC-Scale (Yang et al., 2024) | 82.7 | - |
| SkeletonX (Zhang et al., 2025) | 83.2 | 38.3 |
| **SkelHCC (Ours)** | **84.1** | **40.0** |

100-class setting. CrossGLG (Yan et al., 2024) is the current SOTA multimodal OSAR method, and our approach comprehensively outperforms it (with improvements ranging from 4.1% to 9.0%) across all evaluation settings. Unlike prior methods that fine-tune the entire backbone during one-shot learning, our approach keeps the backbone frozen and introduces only 0.5M trainable parameters, resulting in significantly lower adaptation complexity. The linguistic prior injection of GAP and CrossGLG both require retraining of the backbone, while our SkelHCC only requires the negligible training parameters for EH-HCLIP alignment.

Table 2 compares our method with state-of-the-arts on NTU60 and P-MMD dataset. The results demonstrate that our proposed SkelHCC still outperforms all SOTA methods on NTU60, surpassing CrossGLG by a significant margin of 8.5% in recognition accuracy under the 50-class setting. Furthermore, our method also achieves state-of-the-art performance on the more challenging P-MMD dataset, once again validating its superiority and strong generalization capability. Additional results are provided in the *Appendix* G.

### 4.4. Ablation Studies

Below, we conducted ablation studies to demonstrate the effectiveness and compatibility of the proposed method.

**Baseline.** As shown in Table 3, to ensure a fair compari-

*Table 3.* Validation of the effectiveness of SkelHCC and Internal Components on NTU RGB+D 120 dataset (100 base classes). HCLIP denotes the vanilla hyperbolic CLIP (Desai et al., 2023).

| Method | Acc (%) |
|---|---|
| CLIP (Euclidean)+ Cache | 62.9 |
| HCLIP + Cache | $64.8^{+1.9}$ |
| EH-HCLIP + Cache | $67.6^{+4.7}$ |
| CLIP (Euclidean) + LMV-Cache | $66.2^{+3.3}$ |
| HCLIP + LMV-Cache | $68.2^{+5.3}$ |
| **EH-HCLIP+LMV-Cache (Our SkelHCC)** | $\mathbf{71.6}^{+8.7}$ |

*Table 4.* Comparison of the recognition accuracy of multi-granularity features on NTU RGB+D 120 dataset (100 base classes). BP and BJ denote the body-part level and body-joint level.

| Method | Acc (%) |
|---|---|
| SkelHCC | 71.6 |
| -LMV-Cache $w/o$ BP | $68.4^{-3.2}$ |
| -LMV-Cache $w/o$ BJ | $68.9^{-2.7}$ |
| -LMV-Cache $w/o$ BP, BJ | $67.6^{-4.0}$ |
| -EH-HCLIP $w/o$ BP | $68.8^{-2.8}$ |
| -EH-HCLIP $w/o$ BJ | $69.1^{-2.5}$ |
| -EH-HCLIP $w/o$ BP, BJ | $68.2^{-3.4}$ |

*Table 5.* The impact of different types of masks in our proposed LMV-Cache module on NTU RGB+D 120 dataset (100 base classes).

| Method | Acc (%) |
|---|---|
| Our SkelHCC $w/o$ Mask | 68.5 |
| + Random Mask | $66.3^{-2.2}$ |
| + Learnable Mask | $68.6^{+0.1}$ |
| + Attention Mask | $67.1^{-1.4}$ |
| + LLM Mask $\mathcal{M}_{support}^{BP}$ | $69.9^{+1.4}$ |
| + LLM Mask $\mathcal{M}_{support}^{BP}, \mathcal{M}_{support}^{BJ}$ | $\mathbf{71.6}^{+3.1}$ |

son, we constructed a simple baseline model using CLIP with a training-free cache that stores skeleton features to achieve simple similarity-based retrieval. Specifically, since CLIP is designed for images, we add an MLP interface to the skeleton backbone (Chen et al., 2021; Zhou et al., 2024) and train it on the *base* class dataset to create a CLIP-aligned space. In practice, this training process required only 5 epochs. This simple baseline achieves competitive performance, already outperforming the SOTA multimodal method CrossGLG (Yan et al., 2024).

This superiority can be attributed to the effective incorporation of CLIP-Cache based multimodal priors, which provide a more robust semantic guidance for one-shot retrieval. This fully demonstrates the effectiveness of our basic framework. All experiments follow consistent evaluation protocols with the previous methods to ensure fairness.

**Effectiveness of Internal Components in SkelHCC.** We first validate the effectiveness of the vanilla hyperbolic CLIP (Desai et al., 2023) (HCLIP), which achieves a good improvement (+1.9%) over baseline. We then demonstrate the effectiveness of the proposed EH-HCLIP , which achieves a solid 4.7% performance improvement than baseline. In addition, we further demonstrate the effectiveness of the proposed LMV-Cache, which is specifically designed for skeleton data, we configure it as an adapter to CLIP. As shown in Table 3, this specialization yields a performance gain of 3.3% over a standard cache. Finally, we leverage the LMV-cache to play an adapter for EH-HCLIP. The results demonstrate notable compatibility, leading to a significant improvement (+8.7%) over baseline and outperforming CrossGLG by a significant margin of 9.0%.

**Multi-granularity Human Skeleton Partitioning.** Table 4 validates the effectiveness and necessity of the hierarchical structure prior. BJ and BP denote the body-joint and body-part granularities, which enable the model to consider local features. Both the LMV-Cache and EH-HCLIP integrate this multi-granularity partitioning. Notably, performance drops significantly—by 3.7% and 4.3%, respectively—when EH-HCLIP and LMV-Cache rely solely on full-body features. This confirms that our multi-granularity approach is crucial for learning a robust multi-modal space. The sharper decline in LMV-Cache can be partly attributed to the loss of

guidance from the LLM-guided Mask.

**Mask Selection for the LMV-Cache.** As shown in Table 5, we compare the impact of several mask types on recognition performance. The Random Mask, which assigns random importance scores, leads to a 2.2% performance drop—because the model perceives it as noise. The Learnable Mask, initialized to 1.0, is optimized using a cross-entropy loss applied to the cache logits (since no gradients propagate in the LMV-Cache), yet it brings only negligible improvement. The Attention Mask employs a self-attention mechanism over features and is trained similarly to the learnable mask, but it causes a 1.4% performance degradation. We attribute this to the amplification of semantic bias in the features through self-attention. Finally, we validate the effectiveness of the proposed LLM-based Mask: the body-part mask alone brings a 1.4% performance gain, and adding the body-joint mask further improves gains to 3.1%.

In addition, we compares the impact of different backbones on our proposed SkelHCC. The results demonstrate that our method exhibits excellent compatibility across multiple backbones. Notably, under the same backbone setting, our method outperforms CrossGLG (Yan et al., 2024) by a margin of 11.1%. For further details and additional results, please refer to the Appendix A.

## 5. Conclusion

This work proposed SkelHCC, a novel skeleton cache adaptation framework under hyperbolic CLIP space for achieving powerful one-shot skeleton-based action recognition. Skel-HCC integrates the EH-HCLIP that leverages human struc-

tural priors to perform hierarchical hyperbolic contrastive learning, providing richer linguistic supervision. Additionally, SkelHCC introduces a training-free LMV-Cache, aimed at enhancing EH-HCLIP's one-shot adaptability. It leverages structural priors and high-level knowledge from LLMs to deliver multi-scale, semantically guided predictions from a single exemplar. Our method is evaluated under the standard one-shot action recognition protocol, which better reflects its potential in data-scarce scenarios. Future work will extend to few-shot settings. While SkelHCC includes specialized designs for skeleton data, other modalities such as videos and depth maps are also promising directions for future exploration.

## Acknowledgments

This research was supported by the Australian Government through the Australian Research Council's DECRA funding scheme (Grant No.: DE250100030) and Discovery Project funding schemes (Grant No.: DP260100218, DP260101891). Dr Qiuhong Ke is the recipient of an Australian Research Council Discovery Early Career Researcher Award (project number DE250100030) funded by the Australian Government. This research was also supported by National Natural Science Foundation of China (Grant No.62162068, 62462066, 62466060), the China Scholarship Council (Grant No. 202407030035), Yunnan Province Ten Thousand Talents Program and Yunling Scholars Special Project (Grant No. YNWR-YLXZ-2018-022), Joint Fund Project for "Double First-Class" Construction of Science and Technology Department of Yunnan Province and Yunnan University (Grant No. 202301BF070001-025), Scientific Research Fund of Yunnan Provincial Department of Education (No.2026Y0185, 2025J0008), supported by No. FWCY-BSPY2024009, No. KC-252513122.

## Impact Statement

This paper presents work whose goal is to advance the field of machine learning. There are many potential societal consequences of our work, none of which we feel must be specifically highlighted here.

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

# Appendix Contents

## A. The impact of different backbones on SkelHCC

Table 6 compares the impact of different backbones on our proposed SkelHCC. We selected four authoritative backbones from fully-supervised skeleton action recognition. Our SkelHCC achieves notable recognition performance with all four backbones, with CTR-GCN delivering the best results and thus being selected as the backbone in our experiments. Notably, we compared our method with CrossGLG under the same backbone (HD-GCN), where our approach significantly outperforms it by 11.1%. This demonstrates that the performance advantage of our method over the SOTA is not solely attributable to the use of a stronger backbone.

*Table 6.* Comparison of one-shot skeleton action recognition performance on NTU RGB+D 120 dataset (100 base classes) under different backbones. Params denotes the number of trainable parameters.

| Backbone | Params | Acc (%) |
|---|---|---|
| HD-GCN (Lee et al., 2023) + CrossGLG | 1.8M | 56.8 |
| HD-GCN (Lee et al., 2023) + SkelHCC | 0.5M | $67.9^{+11.1}$ |
| Block-GCN (Zhou et al., 2024) + SkelHCC | 0.5M | 68.1 |
| Shift-GCN (Lee et al., 2023) + SkelHCC | 0.5M | 69.4 |
| **CTR-GCN (Chen et al., 2021) + SkelHCC** | **0.5M** | **71.6** |

## B. Hyperbolic Preliminaries

**Tangent Space and Exponential Map.** Conventional vector operations are not directly applicable in hyperbolic space. Operations are typically conducted in the tangent space (Desai et al., 2023)—a local linear approximation of the hyperbolic manifold—and projected onto the hyperbolic manifold via the exponential map. Specifically, the tangent space $T_{\mathbf{p}}\mathbb{L}^d = \{\mathbf{v} \in \mathbb{R}^{n+1} : \langle \mathbf{p}, \mathbf{v} \rangle_{\mathbb{L}} = 0\}$ at a point $\mathbf{p}$ in hyperbolic space is the set of vectors $\mathbf{v} \in \mathbb{R}^{n+1}$ tangent to the hyperboloid at that point. Next, the projection from tangent space onto hyperbolic manifold is achieved by exponential mapping $\mathrm{expM}(\cdot)$ (Khrulkov et al., 2020) as:

$$\mathbf{x} = \mathrm{expM}_{\mathbf{p}}(\mathbf{v}) = \cosh\left(\sqrt{c}\|\mathbf{v}\|_{\mathbb{L}}\right)\mathbf{p} + \frac{\sinh\left(\sqrt{c}\|\mathbf{v}\|_{\mathbb{L}}\right)}{\sqrt{c}\|\mathbf{v}\|_{\mathbb{L}}}, \quad (12)$$

where c denotes the curvature and $\|\mathbf{v}\|_{\mathbb{L}} = \sqrt{|\langle \mathbf{v}, \mathbf{v} \rangle_{\mathbb{L}}|}$ denotes the *Lorentzian* norm. In addition, the inverse projection is achieved via the logarithmic map. In our work, $\mathbf{p}$ is defined as the origin of the hyperboloid.

Crucially, the exponential mapping at the origin preserves the radial direction, establishing a direct link between the Euclidean norm in the tangent space and the hierarchical depth in the hyperbolic manifold. As indicated in Eq. (12), the hyperbolic distance from the origin is strictly monotonic with respect to the tangent norm $\|\mathbf{v}\|_{\mathbb{L}}$. This property allows

the learned projection head (i.e., the MLP in tangent space) to act as a "hierarchy shaper": it learns to assign smaller norms to generic features (e.g., body granularity, mapping near the origin) and larger norms to fine-grained features (e.g., joint granularity, mapping near the boundary), thereby explicitly capturing the intrinsic hierarchical geometry of the skeleton data.

## C. Body Part Partitioning based on Human Bio-anatomical Priors

As mentioned in Sec. 3.3, we define three granularity levels: 1) *Body Joint* (BJ), 2) *Body Part* (BP), and 3) *Full Body* (FB). BJ ($C \times V$) represents joint-level spatial features, which are obtained by temporally averaging the high-level graph embeddings ($C \times T \times V$) generated by the backbone. FB ($C$) denotes global features, which are obtained by performing global average pooling over both the temporal and spatial dimensions of the skeletal graph representations. FB captures global features, while BJ can measure joint-level fine-grained features. Considering that action execution relies on the collaboration of adjacent joints, we define an intermediate granularity BP ($C \times V_p$) that partitions joints into four body parts according to bio-anatomical priors: head, arms & hands, torso, and legs & feet, where $V_p = 4$. The joint features within each part are averaged. Specifically, the correspondence between each body part and body joint is illustrated in the Fig. 3 (the human joint distribution is consistent across NTU60, NTU120 and P-MMD datasets).

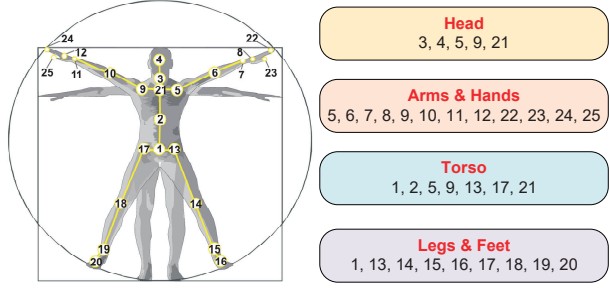

*Figure 3.* The diagram of body part partitioning based on human bio-anatomical priors on NTU60 (Shahroudy et al., 2016), NTU120 (Liu et al., 2020a) and P-MMD (Liu et al., 2020b) action dataset.

## D. Explicit Hierarchical Hyperbolic Entailment Learning

As mentioned in Eq. 6, we introduce a Hyperbolic Entailment Learning (HEL) loss $\mathcal{L}_{HEL}$ followed by (Desai et al., 2023; Ganea et al., 2018b), which leverages the geometric constraints of entailment cones to enforce the partial order (Vendrov et al., 2015) between the paired skeleton

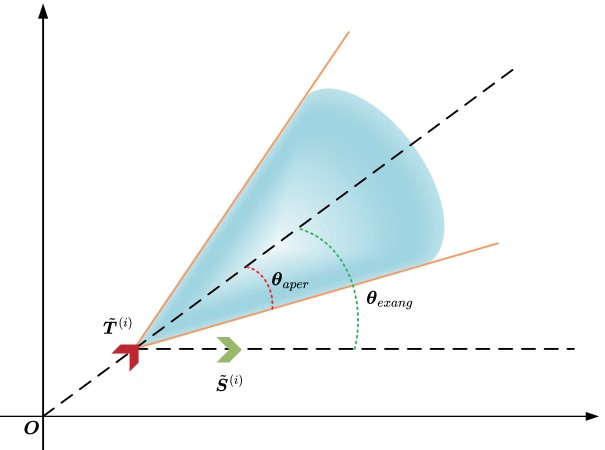

*Figure 4.* The diagram of the entailment cone in hyperbolic space. The exterior angle $\boldsymbol{\theta}_{exang}$ of the skeleton hyperbolic embedding $\tilde{\boldsymbol{S}}^{(i)}$ is pushed within the half-aperture $\boldsymbol{\theta}_{aper}$ of the text hyperbolic embedding $\tilde{\boldsymbol{T}}^{(i)}$.

and text. Here, we provide more details about $\mathcal{L}_{HEL}$. We begin by defining the entailment cone of text hyperbolic embedding $\tilde{\boldsymbol{T}}^{(i)}, i \in \{BJ, BP, FB\}$ (as illustrated in Fig. 4), with its half-aperture $\boldsymbol{\theta}_{aper}$ described as follows:

$$\boldsymbol{\theta}_{aper}(\tilde{\boldsymbol{T}}^{(i)}) = \arcsin\left(\frac{2K}{1 + \sqrt{c}\|\tilde{\boldsymbol{T}}^{(i)}\|^2}\right), \quad (13)$$

where $K = 0.1$ is used to establish boundary conditions. $\|\cdot\|$ denotes the the Euclidean norm. $c$ denotes the curvature. $\boldsymbol{\theta}_{aper}$ is marked with a red dashed line in the Fig. 4. Next, our motivation is to leverage entailment cones to impose geometric constraints on the hyperbolic embeddings of both the skeleton and text, enabling better alignment. Therefore, we further define an exterior angle $\boldsymbol{\theta}_{exang}$, marked with a green dashed line in the Fig. 4, which can be formulated as:

$$\boldsymbol{\theta}_{exang}(\tilde{\boldsymbol{T}}^{(i)}, \tilde{\boldsymbol{S}}^{(i)}) = \arccos(\frac{\tilde{S}_0^{(i)} + \tilde{T}_0^{(i)} c\langle\tilde{\boldsymbol{T}}^{(i)}, \tilde{\boldsymbol{S}}^{(i)}\rangle_{\mathbb{L}}}{\|\tilde{\boldsymbol{T}}^{(i)}\|\sqrt{(c\langle\tilde{\boldsymbol{T}}^{(i)}, \tilde{\boldsymbol{S}}^{(i)}\rangle_{\mathbb{L}})^2 - 1}}),$$

$$\tilde{S}_0^{(i)} = \sqrt{\frac{1}{c} + \|\tilde{\boldsymbol{S}}^{(i)}\|^2},$$

$$\tilde{T}_0^{(i)} = \sqrt{\frac{1}{c} + \|\tilde{\boldsymbol{T}}^{(i)}\|^2},$$

$$(14)$$

where $\tilde{S}_0^{(i)}$ and $\tilde{T}_0^{(i)}$ denote the time component of skeleton hyperbolic embedding $\tilde{\boldsymbol{S}}^{(i)}, i \in \{BJ, BP, FB\}$ and text hyperbolic embedding $\tilde{\boldsymbol{T}}^{(i)}$, respectively. $c$ denotes the curvature. $\|\cdot\|$ denotes the the Euclidean norm. Finally, we have defined the hyperbolic entailment learning (HEL) loss $\mathcal{L}_{HEL}$, which helps push $\tilde{\boldsymbol{S}}^{(i)}$ into the entailment cone

*Table 7.* Comparison of the performance with state-of-the-arts on NTU60 dataset under different base classes settings. **Bold** highlights the best performance. Underline indicates the second-highest performance.

| Base Classes | 10 | 20 | 30 | 40 | 50 |
|---|---|---|---|---|---|
| uDTW (Wang & Koniusz, 2022b) | 56.9 | 61.2 | 64.8 | 68.3 | 72.4 |
| MotionBERT (Zhu et al., 2023b) | 58.3 | 61.0 | 70.0 | 70.3 | 74.5 |
| CrossGLG (Yan et al., 2024) | 57.9 | 67.1 | 70.9 | 73.4 | 75.6 |
| **SkelHCC (Ours)** | **63.3** | **68.9** | **76.5** | **77.4** | **84.1** |

of $\tilde{\boldsymbol{T}}^{(i)}$ by penalizing the excessively large exterior angle $\boldsymbol{\theta}_{exang}$ , formulated as follows:

$$\mathcal{L}_{HEL}(\tilde{\boldsymbol{S}}^{(i)}, \tilde{\boldsymbol{T}}^{(i)}) = \max(0, \boldsymbol{\theta}_{exang}(\tilde{\boldsymbol{T}}^{(i)}, \tilde{\boldsymbol{S}}^{(i)}) - \boldsymbol{\theta}_{aper}(\tilde{\boldsymbol{T}}^{(i)})). \quad (15)$$

Based on this, we construct an Explicit Hierarchical Hyperbolic Entailment (EHHE) loss $\mathcal{L}_{EHHE}$ (in Eq. 6) as

$$\mathcal{L}_{EHHE} = \sum_{i=1}^{|\tilde{\boldsymbol{S}}|} \alpha_i \mathcal{L}_{HEL}(\tilde{\boldsymbol{S}}^{(i)}, \tilde{\boldsymbol{T}}^{(i)}), \quad (16)$$

where $|\tilde{\boldsymbol{S}}|$ denotes the number of elements in the skeleton hyperbolic embedding set $\tilde{\boldsymbol{S}}$. $\alpha_i$ represents learnable weights to dynamically adjust the impact of three granularities.

## E. Details of the action text description

This section provides a more detailed supplement to the action descriptions used in Section 3.3. For full-body descriptions, we designed a prompt template: 'a skeleton video for the action of [*label*]'. For generating body-part and body-joint text features, we created specialized prompts (as illustrated in the Fig. 5) that leverage the powerful reasoning capabilities of LLMs to generate textual descriptions. As shown in Tab. 8, we also provide several examples of body-part descriptions to facilitate intuitive understanding. It is worth noting that for body-joint descriptions, considering that some joint descriptions are overly detailed and personalized - which may introduce noise and affect generalization - we expand full-body text features to optimize them.

## F. Detailed Body-joint and Body-part Mask Prompts

As shown in Fig. 6, we provide the detailed prompts as mentioned in (Fig. 2 in Paper) for generating masks using the LLM (GPT 4). Moreover, we provide some examples of body-part masks to facilitate a more intuitive understanding. The first 30 action categories from the NTU120 dataset were selected for this purpose, as shown in Tab. 9. Taking "A23. hand waving" as an example, the weight on arms & hands is 0.75 while weights on other body parts are insignificant, which fully aligns with the action semantics of hand waving. Similarly, for "A27. jump up", the weight on leg & feet is 0.6 and the weight on arms & hands is 0.15, indicating that

*Table 8.* LLM-generated body-part action descriptions. The first 10 categories from the NTU (Shahroudy et al., 2016; Liu et al., 2020a) dataset are selected as examples.

| Action Classes | head | arms&hands | torso | leg&feet |
|---|---|---|---|---|
| A1. drink water | 'Tilt back to drink' | 'Grasp the bottle or cup, lift it up to mouth.' | 'Lean forward to reach the cup or glass' | 'Remain still to support the body' |
| A2. eat meal/snack | 'Chew and swallow food.' | 'Hold utensils and bring food to the mouth.' | 'Support the body while eating.' | 'Remain still while seated at the table.' |
| A3. brushing teeth | 'Tilt back slightly' | 'Hold toothbrush and move it in circular motions' | 'Remain still' | 'Remain still' |
| A4. brushing hair | 'Move side to side to brush hair.' | 'Grip the brush and move it through the hair.' | 'Remain still to provide stability.' | 'Remain still to provide stability.' |
| A5. drop | 'Lower quickly' | 'Lower quickly' | 'lower quickly' | 'Bend slightly' |
| A6. pickup | 'Look at the object to be picked up.' | 'Reach out and grasp the object.' | 'Bend forward slightly to reach the object.' | 'Remain stationary to provide a stable base for the movement.' |
| A7. throw | 'Tilt back slightly' | 'Grip the object and propel it forward' | 'Twist and rotate to generate momentum' | 'Push off the ground to generate power' |
| A8. sitting down | 'Lower slowly' | 'Reach out to support the body' | 'Bend at the hips and lower down' | 'Bend at the knees and lower to the ground' |
| A9. standing up | 'Lift up and straighten.' | 'Push off the chair arms or surface to help lift the body.' | 'Straighten and lift up.' | 'Straighten and bear weight to stand.' |
| A10. clapping | 'Move up and down slightly.' | 'Come together quickly and sharply.' | 'Remain still.' | Remain still.' |

arms and hands naturally swing during the jump up motion, again completely consistent with the semantics.

Interestingly, the action "A20. Put on a hat" often involves fine-grained adjustments (e.g., aligning the hat, checking fit), requiring significant engagement of the head alongside the hands. The LLM's equal weighting (0.40/0.40) captures this interaction. The action "A21.Take off a hat" typically involves a dominant grasping and lifting motion by the Arms, with the head remaining relatively passive. The LLM's shift towards Arm dominance (0.55) and reduced Head weight (0.25) essentially reflects this "Agent (Hand) vs. Object (Head)" relationship. Thus, the LLM captures the semantic intent of the motion rather than just rigid physical symmetry.

More examples of body-joint and body-part masks are stored in CSV tables and submitted as supplementary materials.

## G. Additional Experimental Results on NTU-60

Tab. 7 compares several state-of-the-art methods under configurations of 10 to 50 base classes to demonstrate the superiority of our approach. Our proposed SkelHCC significantly outperforms all these SOTA methods across every evaluation metric. Compared to the representative method MotionBERT, our approach achieves a notable improvement of 9.6% under the 50 base-class setting. Importantly, when compared to the SOTA multimodal method CrossGLG, our method yields substantial gains of 5.4% and 8.5% under the 10 and 50 base-class settings, respectively.

## H. Sensitivity Analysis of LLM-generated Masks

In this section, we are dedicated to conducting sensitivity analysis of Large Language Model (LLM) based masks.

Firstly, due to the stochastic nature of the generated masks may vary across different inference sessions. To evaluate the stability of our methods, we generated 10 independent sets of masks using GPT-4. The performance distribution after integration with our SkelHCC is shown in the Fig 7 (blue box plots). Specifically, the results show a narrow distribution with a minimal standard deviation (e.g., $\sigma < 0.2\%$), indicating that our SkelHCC is robust to minor fluctuations in the LLM-generated masks. It is worth noting that even the minimum accuracy among the 10 runs (the bottom whisker) significantly outperforms the SkelHCC $w/o$ Mask (gray dashed line) by a large margin. This confirms that the gains of incorporating LLM knowledge are consistent and reliable, not accidental.

To further mitigate the potential noise in single-run LLM generation, we leverage the mean LLM mask strategy (10 runs), where we average the mask weights from multiple LLM queries to form a consensus action semantic prior.

As shown by the orange stars in Fig 7, the model trained with the mean LLM mask consistently achieves superior performance compared to the median accuracy of individual runs, which suggests that the ensemble of semantic knowledge effectively filters out random noise and captures more accurate body-joint/body-part correlations, making it the optimal solution for our methods.

*Table 9.* Example of body part mask output by LLM. We have selected the first 30 categories from NTU120 (Liu et al., 2020a). LLM-generated body-part prior weights representing relative perceptual importance of body parts for each action class. As the priors capture visual saliency rather than physical symmetry, logically inverse actions (e.g., "put on a hat" vs. "take off a hat") are not constrained to share identical weights.

| Action Classes | head | arms&hands | torso | leg&feet |
|---|---|---|---|---|
| A1. drink water | 0.30 | 0.50 | 0.10 | 0.10 |
| A2. eat meal/snack | 0.30 | 0.45 | 0.15 | 0.10 |
| A3. brushing teeth | 0.20 | 0.60 | 0.15 | 0.05 |
| A4. brushing hair | 0.20 | 0.60 | 0.10 | 0.10 |
| A5. drop | 0.05 | 0.25 | 0.20 | 0.50 |
| A6. pickup | 0.10 | 0.55 | 0.15 | 0.20 |
| A7. throw | 0.10 | 0.60 | 0.15 | 0.15 |
| A8. sitting down | 0.10 | 0.15 | 0.20 | 0.55 |
| A9. standing up (from sitting position) | 0.05 | 0.15 | 0.20 | 0.60 |
| A10. clapping | 0.05 | 0.75 | 0.10 | 0.10 |
| A11. reading | 0.35 | 0.50 | 0.10 | 0.05 |
| A12. writing | 0.10 | 0.70 | 0.10 | 0.10 |
| A13. tear up paper | 0.10 | 0.70 | 0.10 | 0.10 |
| A14. wear jacket | 0.10 | 0.60 | 0.20 | 0.10 |
| A15. take off jacket | 0.10 | 0.60 | 0.20 | 0.10 |
| A16. wear a shoe | 0.05 | 0.25 | 0.10 | 0.60 |
| A17. take off a shoe | 0.10 | 0.50 | 0.10 | 0.30 |
| A18. wear on glasses | 0.65 | 0.25 | 0.05 | 0.05 |
| A19. take off glasses | 0.35 | 0.40 | 0.10 | 0.15 |
| A20. put on a hat/cap | 0.40 | 0.40 | 0.10 | 0.10 |
| A21. take off a hat/cap | 0.25 | 0.55 | 0.10 | 0.10 |
| A22. cheer up | 0.15 | 0.45 | 0.20 | 0.20 |
| A23. hand waving | 0.10 | 0.75 | 0.05 | 0.10 |
| A24. kicking something | 0.05 | 0.10 | 0.20 | 0.65 |
| A25. reach into pocket | 0.05 | 0.60 | 0.15 | 0.20 |
| A26. hopping (one foot jumping) | 0.05 | 0.15 | 0.10 | 0.70 |
| A27. jump up | 0.10 | 0.15 | 0.15 | 0.60 |
| A28. make a phone call/answer phone | 0.15 | 0.65 | 0.10 | 0.10 |
| A29. playing with phone/tablet | 0.20 | 0.60 | 0.10 | 0.10 |
| A30. typing on a keyboard | 0.15 | 0.65 | 0.10 | 0.10 |

*Table 10.* **Sensitivity analysis of different LLM backbones.** Accuracy comparison of 20 and 100 base classes on the NTU120 dataset.

| Method / LLM Backbone | 20 class (%) | 100 class (%) |
|---|---|---|
| *CrossGLG*(Yan et al., 2024) | *45.3* | *62.6* |
| Our SkelHCC $w/o$ Mask | 49.9 | 68.5 |
| $w/$ LLM (Deepseek-v3) Mask | 51.7 | **71.6** |
| $w/$ LLM (Qwen-3) Mask | 51.9 | 71.5 |
| $w/$ **LLM (GPT-4) Mask** | **52.0** | **71.6** |

Furthermore, we investigated the sensitivity of our framework to different LLM backbones, including open-source models like Deepseek-v3 and Qwen-3, as compared to GPT-4 in Table 10. Quantitatively, all LLM-enhanced variants significantly outperform the SkelHCC w/o Mask baseline

(e.g., improving from 49.9% to ∼52.0% on the 20-class setting), validating the crucial role of semantic priors. Notably, the performance variance among different LLMs is marginal. For instance, Qwen-3 achieves 71.5% accuracy on the 100-class setting, which is comparable to the 71.6% achieved by GPT-4. This consistency indicates that the effectiveness of our framework is model-agnostic and stems from the integration of structured semantic knowledge rather than the specific capability of a single LLM.

# I. Adaptability Analysis of Hyperbolic Space and skeleton data

Recent advanced methods (Franco et al., 2023; Li et al., 2025) have demonstrated the intrinsic adaptability between skeleton data and hyperbolic space.

**◎ Body-part action descriptions prompt**

> I want to build a classification model that strengthens action recognition by enhancing local-level feature detection. You need to generate, for the action of **{{action}}**, a list of descriptive sentences in which each sentence describes the local motion that a specific body part area needs to complete when a person is performing the action. The description does not need to describe the purpose of the motion, just output the simplest and most common phrase of the motion. The description for each body part area  head, arms & hands, torso and legs & feet. Output format should be a two-column table whose column names are relatively the name of each body joint area and its local motion description with a format of "desc: " at the front. Don't polish the output, just output the table. Don't output the column title.

**◎ Body-joint action descriptions prompt**

> I want to build a classification model that strengthens action recognition by enhancing local-level feature detection. You need to generate, for the action of **{{action}}**, a list of descriptive sentences in which each sentence describes the local motion that a specific body joint area needs to complete when a person is performing the action. The description does not need to describe the purpose of the motion, just output the simplest and most common phrase of the motion. The description for each body joint area should be provided in the following order: base of spine, middle of spine, neck, head, left shoulder, left elbow, left wrist, left hand, right shoulder, right elbow, right wrist, right hand, left hip, left knee, left ankle, left foot, right hip, right knee, right ankle, right foot, spine, tip of left hand, left thumb, tip of right hand, right thumb. Output format should be a two-column table whose column names are relatively the name of each body joint area and its local motion description with a format of "desc: " at the front. Don't polish the output, just output the table. Don't output the column title.

**Full-body (global) action prompt**

> "a skeleton video for the action of **{{action}}**"

*Figure 5.* Detailed action descriptions Prompts.

**◎ Body-part mask prompt**

> I want to build a classification model that strengthens action recognition by enhancing local-level feature judgment. You need to generate, for given actions, a list that provides the representativeness (degree of contribution) of each body part when recognizing the action, in percentages (0.0–1.0). The sum of all body-part importance should equal 1.0. There are 4 body parts for total. Each body part's importance should be provided in the following order: head, arms & hands, torso and legs & feet. Output format should be a two-column table whose column names are relatively the name of each body part name and its importance percentage (0.0–1.0). Don't polish the output, just output the table. Don't output the column title.

**◎ Body-joint mask prompt**

> I want to build a classification model that strengthens action recognition by enhancing local-level feature judgment. You need to generate, for given actions, a list that provides the representativeness (degree of contribution) of each body part when recognizing the action, in percentages (0.0–1.0). The sum of all body-part importance should equal 1.0. Each body part's importance should be provided in the following order: base of spine, middle of spine, neck, head, left shoulder, left elbow, left wrist, left hand, right shoulder, right elbow, right wrist, right hand, left hip, left knee, left ankle, left foot, right hip, right knee, right ankle, right foot, spine, tip of left hand, left thumb, tip of right hand, right thumb. Output format should be a two-column table whose column names are relatively the name of each body part name and its importance percentage (0.0–1.0). Don't polish the output, just output the table. Don't output the column title.

*Figure 6.* Detailed Body-joint and Body-part Mask Prompts.

*Table 11.* Comparison of intrinsic geometry. Lower $\delta_{\text{rel}}$ indicates better hierarchical structure. FB, BP and BJ mean Full-Body, Body-Part and Body-Joint, respectively.

| Feature Level | Relative $\delta$ ($\delta_{\text{rel}}$) |
|---|---|
| Full-Body Only | 0.0194 |
| Body-Joint Only | 0.0171 |
| **Ours (FB+BP+BJ)** | **0.0158** |

To justify the incorporation of hyperbolic geometry and validate the effectiveness of our hierarchical methods (Joint → Part → Global), we analyzed the intrinsic geometry of the learned feature space using Gromov's $\delta$-hyperbolicity (Li et al., 2025). This metric quantifies the extent to which a metric space resembles a tree structure, where a lower $\delta$ indicates a higher degree of hyperbolicity (i.e., stricter hierarchy).

For a metric space $(X, d)$, Gromov's four-point condition states that the space is $\delta$-hyperbolic if, for any four points $x, y, z, w \in X$, the following inequality holds:

$$(x, z)_w \geq \min\left((x, y)_w, (y, z)_w\right) - \delta), \qquad (17)$$

where $(x, y)_w$ denotes the Gromov product with respect to a base point $w$, defined as:

$$(x, y)_w = \frac{1}{2}\big(d(x, w) + d(y, w) - d(x, y)\big). \qquad (18)$$

In this work, we report the Relative $\delta$ ($\delta_{\text{rel}}$), defined as the mean $\delta$ ($\delta_{\text{avg}}$) normalized by the diameter ($D$) of the point set: $\delta_{\text{rel}} = \frac{2\delta_{\text{avg}}}{D}$. A value of $\delta_{\text{rel}} < 0.1$ is typically considered strong evidence of intrinsic hierarchical structure suitable for hyperbolic embedding, whereas Euclidean spaces exhibit significantly higher values (Ganea et al., 2018a).

We conducted an additional ablation study to assess the geometric properties of feature embeddings across different levels of granularity. We sampled feature points from the NTU120 dataset and computed $\delta_{\text{rel}}$ under three configurations: 1) Full-Body only: incorporating only global action embedding; 2) Body-Joint only: utilizing raw joint features; and 3) the unified embedding space comprising Full-Body (FB), Body-Part (BP), and Joint (BJ) features. The results are shown in Tab. 11. We observe that the $\delta_{\text{rel}}$ of the standard Full-Body representation is remarkably low, indicating an intrinsic compatibility between skeletal features and hyperbolic space. Furthermore, upon applying our explicit hierarchical strategy, the $\delta_{\text{rel}}$ decreases further to 0.0158, providing a compelling motivation for the adoption of hyperbolic geometry in our method.

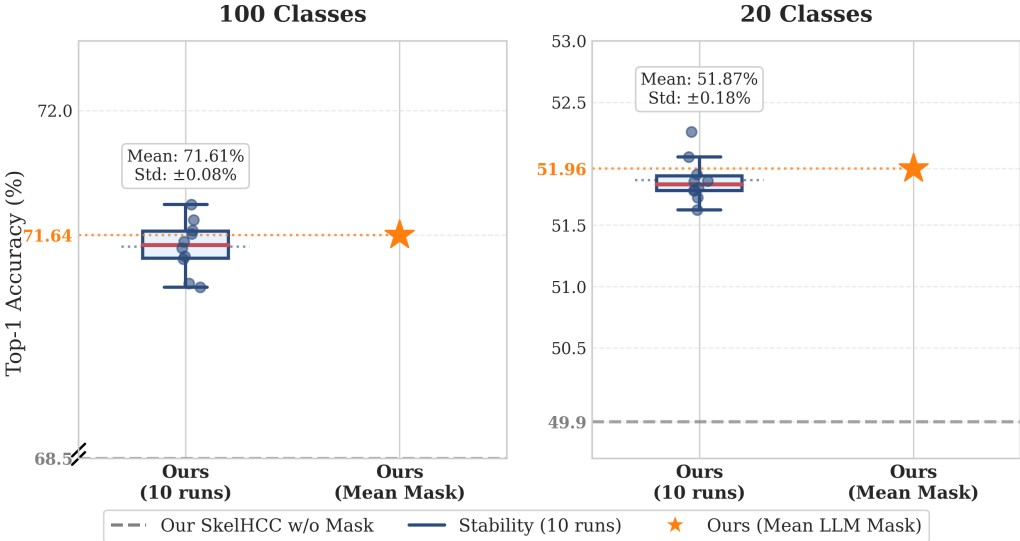

*Figure 7.* **LLM Mask Sensitivity and Stability Analysis.** We compare the SkelHCC w/o mask (gray line), the distribution of 10 independent runs using GPT-4 generated masks (blue box plots), and our used mean LLM mask strategy (orange stars). The results on both 100-class and 20-class settings demonstrate that our method is highly robust to LLM stochasticity, and the mean LLM mask strategy effectively synthesizes semantic knowledge to achieve state-of-the-art performance.

*Table 12.* Complexity and inference efficiency of different cache variants.

| Variant | Acc. (%) | FLOPs / query | FPS |
|---|---|---|---|
| EH-HCLIP | 59.1 | 3.87 G | 202.59 |
| + Cache (FB) | 67.6 | +0.020 M | 187.26 |
| + Cache (FB + BP) | 68.9 | +0.062 M | 179.36 |
| + Cache (FB + BJ) | 68.4 | +0.277 M | 181.43 |
| + LMV-Cache | 71.6 | +0.318 M | 169.49 |

*Table 13.* Deployment overhead of LMV-Cache.

| Process | Peak GPU Mem. (GB) | Time / episode (ms) |
|---|---|---|
| Whole adaptation pipeline | 0.95 | 379.66 |
| Cache construction only | 0.79 | 77.40 |

inference is required during testing. These results indicate that LMV-Cache introduces low deployment overhead while providing consistent performance gains.

## J. Computational Overhead

We further analyze the computational overhead and deployment efficiency of the proposed LMV-Cache. As shown in Table 12, our method introduces only negligible additional computation over EH-HCLIP. The base EH-HCLIP model requires 3.87 GFLOPs per query and achieves 202.59 FPS. Adding the proposed cache introduces only lightweight multi-granularity similarity computations. Even the full LMV-Cache with full-body, body-part, and body-joint cues adds merely 0.318 MFLOPs per query, accounting for approximately 0.008% of the EH-HCLIP computation, while still maintaining real-time inference speed at 169.49 FPS.

We also report the memory usage and episode-level processing time in Table 13. The whole adaptation pipeline requires less than 1 GB GPU memory and takes 379.66 ms per episode. The cache construction stage itself requires only 0.79 GB GPU memory and 77.40 ms per episode. Since LMV-Cache is training-free and the LLM-derived masks are pre-computed offline before deployment, no online LLM

## K. Case study on imperfect LLM priors

We further provide a case-level analysis to examine whether LMV-Cache is robust to imperfect LLM-derived body-part priors. As shown in Table 14, using the BP hierarchy with a uniform mask already brings strong performance on fine-grained actions such as *reach into pocket* and *put on headphone*. However, replacing the uniform BP mask with the LLM-derived BP mask slightly decreases the accuracy, e.g., from 58.10% to 56.30% on *reach into pocket* and from 85.87% to 83.62% on *put on headphone*. This indicates that BP-level LLM priors may be imperfect for some classes.

Importantly, when we introduce the finer BJ hierarchy, even with a uniform BJ mask, the performance is clearly recovered and improved, reaching 59.79% and 88.01% on the two classes, respectively. This suggests that the multi-granularity hierarchy does not simply rely on the correctness of a single LLM prior, but can compensate for noisy BP-level guidance through more localized joint-level evidence. Finally, the full LMV-Cache with both BP and BJ LLM

*Table 14.* Case-level evidence that the hierarchy mitigates imperfect BP-level LLM priors.

| Variant | reach into pocket | put on headphone |
|---|---|---|
| + BP Hierarchy + BP Uniform mask | 58.10 | 85.87 |
| + BP Hierarchy + BP LLM mask | 56.30 (-1.80) | 83.62 (-2.25) |
| + BP, BJ Hierarchy + BP LLM mask + BJ Uniform mask | 59.79 (+3.49) | 88.01 (+4.39) |
| + BP, BJ Hierarchy + BP, BJ LLM masks (LMV-Cache) | 64.87 (+5.08) | 89.29 (+1.28) |

masks achieves the best results, further improving the accuracy to 64.87% and 89.29%. These results demonstrate that the proposed BP/BJ hierarchical design effectively mitigates imperfect BP-level LLM priors and provides more reliable fine-grained action recognition.

