# OpenReview forum: "SkelHCC: A Hyperbolic CLIP-Driven Cache Adaptation Framework for Skeleton-based One-Shot Action Recognition"
_ICML.cc/2026/Conference — ICML 2026 regular_

### Official Review · Reviewer_9vR6 · 2026-03-12

**Soundness:** 2
**Presentation:** 3
**Significance:** 2
**Originality:** 2
**Overall Recommendation:** 3
**Confidence:** 3

**Summary:**

This work proposes a novel skeleton cache adaptation framework, SkelHCC, designed to address representation alignment and adaptability issues in existing skeleton-based action recognition methods. The method leverages structural priors to perform hierarchical hyperbolic contrastive learning, providing richer semantic supervision. It further introduces a training-free cache that utilizes structural priors and the high-level knowledge of large pretrained models to calibrate prediction results. Experimental results show that SkelHCC achieves state-of-the-art performance compared with other methods, highlighting its potential in data-scarce scenarios.

**Compliance With Llm Reviewing Policy:**

Affirmed.

**Final Justification:**

The authors have addressed most of my concerns. However, I remain skeptical about the computational overhead introduced by LLMs, whether through pre-computation or real-time inference, as it may present challenges for practical deployment. Consequently, I have reservations about the algorithm’s real-world applicability.

**Key Questions For Authors:**

See Strength and weakness

**Limitations:**

See Strength and weakness

**Strengths And Weaknesses:**

Strength
1，	Based on the strong compatibility between skeletal data and hyperbolic geometry, this paper introduces an explicit hierarchical hyperbolic CLIP module that embeds skeletal sequences and action language into a shared hyperbolic space to generate structurally consistent representations.
2，	The paper further introduces a training-free LMVCache, which performs adaptive adjustment of the learned hyperbolic space during testing in order to refine prediction results.
3，	The authors validate the effectiveness of the proposed method on datasets such as NTU RGB+D 60 and NTU RGB+D 120, demonstrating its superior performance.

Weakness
1，	The paper mentions the representation alignment problem; however, hyperbolic geometric modeling has already been explored in prior works such as [1] and [2]. Regarding the adaptability issue, cache-based zero-shot test-time adaptation methods have also been proposed in Skeleton-Cache [3]. Although the proposed method achieves promising performance, the authors should further clarify how their approach differs from these existing methods.
[1] Hyperbolic linear attention for skeleton-based human action recognition
[2] Informative Sample Selection Model for Skeleton-based Action Recognition with Limited Training Samples
[3] Boosting Skeleton-based Zero-Shot Action Recognition with Training-Free Test-Time Adaptation
2，	Table 1 reports the number of optimized parameters, but it lacks comparisons of the memory cost required during the model adaptation process as well as the time required to adapt a single sample, particularly considering the introduction of LLMs and cache mechanisms. These factors represent key bottlenecks when deploying the model in resource-constrained environments. Furthermore, the paper should more clearly explain how the introduction of different modules affects GPU memory consumption and inference time in order to better assess the performance gains brought by each component.
3，	When the action name or description is inaccurate, the LLM may generate incorrect weights.The paper should discuss to what extent such cases may affect the model’s performance.
4，	In addition, the suboptimal result annotation in Table 2 appears to contain an error.

---

> ### Author Rebuttal · Authors · 2026-03-31
>
> **[Hyperbolic and Adaptability]** We thank the reviewer for pointing out these relevant works. We agree that both hyperbolic geometry and adaptation have been explored in skeleton understanding.  Our claim is therefore not that SkelHCC is the first skeleton method to use hyperbolic modeling or adaptation in isolation. Rather, our contribution is a unified framework for **one-shot SAR** that combines:
> + **explicit hierarchical skeleton–language alignment** in a shared hyperbolic space (EH-HCLIP).
> + **multi-granularity cache inference** on top of it (LMV-Cache).
>
> We clarify the main differences below.
>
>  **(1)  [1,2] neither consider skeleton–language alignment nor address the one-shot setting.**
>
> [1] applies hyperbolic geometry to fully supervised skeleton representation learning, and [2] uses it for sample selection in semi-supervised learning, without cross-modal alignment or one-shot setting. By contrast, we use hyperbolic space for hierarchical skeleton–language alignment, improving over HCLIP by +3.4 and over CLIP by +5.0 (Tab. 3), which suggests that structured cross-modal alignment, rather than hyperbolic modeling alone, drives the gain.
>
> **(2) [3] is the zero-shot SAR method, but its adaptation differs from ours in several key aspects:**
>
> + **Adaptation driver.** [3] uses **test-driven self-adaptation** using pseudo-labeled test samples, while ours is support-driven one-shot adaptation from labeled exemplars.
>
> + **Cache mechanism.** [3] uses an **online self-updating cache** built from test-time predictions, while our LMV-Cache is a **fixed support cache** constructed from exemplars for one-shot retrieval.
>
> + **Role of language.** [3] uses language for spatial and temporal descriptor-wise weighting, while we use it for multi-granularity support–query similarity and hyperbolic skeleton–text retrieval.
>
> **[Adaptation Cost]** We thank the reviewer for this helpful comment.  We will complement with:
>
> (1) **Online query inference:** the novel-class per-query overhead increases only +0.318M FLOPs, while still achieving 169.49 FPS and +12.5% gain.
>
> (2) **Adaptation memory:** the whole pipeline only requires <1 GB GPU memory. The LLM is not invoked online; all masks are precomputed offline.
>
> **Table R1. Online query inference cost.**
> |Variant|Acc|FLOPs/query|FPS|
> |-|-|-|-|
> |EH-HCLIP|59.1|3.87 G|202.59|
> |+ Cache (FB)|67.6 |+0.020 M|187.26|
> |+ Cache (FB + BP)|68.9|+0.062 M|179.36|
> |+ Cache (FB + BJ)|68.4|+0.277 M|181.43|
> |+ LMV-Cache (FB + BJ + BP)|71.6 (+12.5)|+0.318 M|169.49|
>
> **Table R2. Adaptation memory.**
> |Process| Peak GPU Mem (GB)| Time / episode (ms)|
> |-|-|-|
> |Whole adaptation|0.95|379.66|
> |Cache construction only (LMV-Cache)|0.79|77.40|
>
> **[Mask Robustness]** We acknowledge that LLM priors may be imperfect for novel or ambiguous actions. However, our design mitigates this in two ways:
> (1) masks are used only as **soft weights**, not hard constraints (Eq.11);
> (2) the final prediction is based on **multi-granularity fusion (FB/BP/BJ)**, so the model does not rely on a single prior alone.
>
> This robustness is supported by:
>
> - **Ablations (Tab. 5, Tab. R3):** corrupted masks consistently hurt performance (e.g., **71.6 → 68.6** with label-shuffled masks), showing that correct priors help, while imperfect ones do not collapse the model.
> - **Controlled perturbations (Tab. R3):** suppressing **high-weight** regions causes a larger drop than suppressing **low-weight** ones (**71.6 → 68.2** vs. **69.8**), confirming that the masks capture meaningful discriminative regions.
> - **Case studies and failure analysis (Tabs. R4–R5):** Gains are largest for actions with clear body-region cues (e.g., *tear up paper*), while smaller gains or failures occur for subtle or object-dependent actions (e.g., *use a fan*). Notably, when BP-level priors are imperfect, adding BJ-level evidence can recover performance, suggesting a coarse-to-fine compensation mechanism and robustness to noisy LLM priors.
>
> **Table R3. Sanity checks.** *(w/o mask: 68.5)*
> |Method|BJ (k=5)|BJ(k=5)+BP(k=1)|
> |-|-|-|
> |Correct Mask|69.6|71.6|
> |Label-shuffled Mask |67.9|68.6|
> |Top-k Suppression|68.3 ↓1.3|68.2 ↓3.4|
> |Low-k Suppression|69.3 ↓0.3|69.8 ↓1.8|
>
> **Table R4. Gains  with LLM masks/Hierarchy.**
>
> | Method|Tear up paper|Sniff|Open a box|Use a fan|
> |-|-|-|-|-|
> |LMV-Cache w/o BP,BJ Hierarchy|27.21|23.35|13.64|63.07
> |LMV-Cache w/o LLM Mask|30.84|25.03|13.85|63.49|
> |LMV-Cache|36.92 (+6.08)|30.36 (+5.33)|12.59 (-1.26)|60.84 (-2.65)|
>
> **Table R5. Case-level evidence that hierarchy mitigates imperfect BP-level LLM priors.**
> |Variant (FB Cache)|reach into pocket |put on headphone|
> |-|-|-|
> |+ BP Hierarchy + BP Uniform mask|58.10|85.87|
> |+ BP Hierarchy + BP LLM mask|56.30 (-1.80↓)|83.62 (-2.25↓)|
> |+ BP, BJ Hierarchy + BP LLM mask + BJ Uniform mask|59.79 (+3.49↑)|88.01 (+4.39↑)|
> |+ BP, BJ Hierarchy + BP, BJ LLM masks (LMV-Cache)|64.87 (+5.08↑)|89.29 (+1.28↑)|
>
> **[Tab.2]** indeed has a typo. Thank you for pointing this out. We will correct it in the revision.

---

> > ### Author Rebuttal · Reviewer_9vR6 · 2026-04-03
> >
> > The authors have addressed most of my concerns. However, I remain skeptical about the computational overhead introduced by LLMs, whether through pre-computation or real-time inference, as it may present challenges for practical deployment. Consequently, I have reservations about the algorithm’s real-world applicability.

---

> > > ### Author Response · Authors · 2026-04-03
> > >
> > > We sincerely thank the reviewer for raising this important and practically relevant concern. We realize that our current presentation may not have been sufficiently clear and may have given the impression that the LLM contributes to deployment-time overhead. We would like to clarify that this is not the case:
> > >
> > > While LLMs may introduce additional cost during pre-computation, in our framework they are used **only in a one-time offline stage prior to deployment** to generate semantic masks from the action label names of the support classes, not during deployment. This cost is **amortized across all episodes** and **does not scale with the number of test queries**, making it fundamentally different from runtime overhead. As such, the LLM **is not part of the deployed pipeline** and does not contribute to inference latency or per-sample cost.  Therefore, the reported FLOPs, FPS, and memory usage reflect the actual deployment cost. These results indicate that the proposed method is practical for real-world deployment, given its high throughput and modest memory footprint. We will clarify this point explicitly in the revised manuscript.
> > >
> > > **We will release the code** to facilitate reproducibility and to allow practitioners to directly evaluate the computational cost in real-world settings.

---

### Official Review · Reviewer_zRie · 2026-03-12

**Soundness:** 3
**Presentation:** 3
**Significance:** 4
**Originality:** 4
**Overall Recommendation:** 4
**Confidence:** 3

**Summary:**

The paper proposes a method for skeleton-based one-shot action recognition using hyperbolic representation learning. The authors introduce a framework called SkelHCC, which leverages hyperbolic embeddings together with CLIP to learn a hierarchical and text-aligned representation of skeleton motion. The framework consists of two main components: EH-HCLIP and LMV-Cache. The EH-HCLIP module learns skeleton representations in hyperbolic space, enabling the model to capture hierarchical relationships between actions while aligning them with textual semantics through CLIP, while LMV-Cache is designed for test-time adaptation, allowing the model to incorporate information about novel classes during inference. The method is evaluated on several standard skeleton-based action recognition benchmarks, including NTU RGB+D 60, NTU RGB+D 120, and PKU-MMD II, where the authors report competitive results.

**Compliance With Llm Reviewing Policy:**

Affirmed.

**Final Justification:**

The authors have addressed my concerns. One reviewer also raised the computational overhead of using an LLM, which I agree is a valid point. That said, I still believe the paper provides meaningful value to the community, and I will maintain my positive rating.

**Key Questions For Authors:**

Mainly the points mentioned in the weakenss need to be clarified.

**Limitations:**

The authors do not discuss any limitations or societal impact.

**Strengths And Weaknesses:**

### Strengths

**S1.** The idea of encoding skeleton information in hyperbolic space to capture hierarchical structure is interesting and well aligned with the nature of skeleton-based action recognition.

**S2.** The method achieves strong empirical performance across multiple datasets and reports state-of-the-art results consistently.

**S3.** The paper includes a comprehensive ablation study evaluating the contribution of each component, which helps justify the design choices made in the proposed framework.

---

### Weaknesses

**W1.** The approach heavily relies on LLM guidance. Since LLMs contain substantial world knowledge about actions, they can often generate detailed descriptions of tasks and body-part interactions. In particular, the LLM-guided masking mechanism constructs masks based on textual descriptions of body parts, and the ablations suggest this component provides a significant performance boost. However, since the baselines do not use comparable LLM-derived descriptions, it is unclear whether the comparisons are entirely fair.

**W2.** There appears to be a potential issue with Equation 5. Based on the definition of the distance function provided in Equation 3, the distance appears to be symmetric. If that is the case, the two terms used in the loss \(L_{HCL}\) would become identical, which raises questions about whether the formulation is correct or whether some asymmetry was intended.

**W3.** The writing of the method section is quite dense and difficult to follow, making it challenging to clearly understand the pipeline and the interaction between the different components.

---

> ### Author Rebuttal · Authors · 2026-03-31
>
> We thank the Reviewer zRie for the careful and constructive comments. We have carefully considered and addressed each concern point by point in the rebuttal.
>
> **[Fairness]** We thank the reviewer for raising this important concern. We clarify this issue from three aspects.
>
> **(1) No test leakage.**
> The LLM masks in LMV-Cache are generated **only from the action label names** with a predefined BJ/BP vocabulary (Fig. 2), and are used only to reweight BJ/BP similarities during retrieval (Eq. 9). They do not use query labels or any information from novel query samples. The final prediction still depends on the full retrieval score (Eq. 11). Therefore, this is consistent with the one-shot protocol and does not introduce test leakage. Importantly, the mask is a structured body prior from the action name, rather than an LLM action description directly used for recognition.
>
> **(2) Fair controlled comparison.**
> For fair comparison, we fix the backbone, CLIP encoder, retrieval mechanism, and one-shot protocol, and vary **only the mask source**. Under this controlled setting, the LLM mask outperforms all alternatives. Moreover, shuffled-mask and suppression tests show that the gain comes from an **action-consistent structured prior**, rather than simply adding an extra mask.
>
> **Table R1. Fair comparison of our LLM mask with baseline masks.** (k=1 for BP and k=5 for BJ)
>
> | Mask setting  (BJ, BP) | Acc. (%) |
> | ---------------------- | -------: |
> | Uniform Mask           |     68.5 |
> | Random Mask            |     66.3 |
> | Learnable Mask         |     68.6 |
> | Attention Mask         |     67.1 |
> | Label-shuffled Mask    |     68.6 |
> | Top-k Suppression Mask |     68.2 |
> | Low-k Suppression Mask |     69.8 |
> | **LLM Mask**           | **71.6** |
>
> **(3) The gain is not from LLM masks alone.**
> A substantial improvement already comes from the non-mask components. As shown in Tab. R2, moving from the baseline (62.9) to SkelHCC w/o LLM mask (68.5) gives +5.6, showing that explicit hierarchy, hyperbolic alignment, and cache adaptation contribute strongly on their own. The LLM mask then provides an additional improvement from 68.5 to 71.6. Thus, the final gain comes from the ****combination**** of the architectural design and the structured semantic prior, rather than from LLM guidance alone.
>
> **Table R2. Contribution of non-mask design and LLM masks.**
>
> | Method               | Acc. (%) | Gain |
> | -------------------- | -------: | ---: |
> | CLIP + Cache         |     62.9 |    – |
> | SkelHCC w/o LLM mask |     68.5 | +5.6 |
> | SkelHCC              |     71.6 | +8.7 |
>
> Overall, these results show that SkelHCC is a principled OSAR framework, in which the LLM-guided mask is an additional and fairly evaluated component.
>
>
>
> **[Eq. 5 Asymmetry]** Thank you for this careful observation. We agree that the Lorentzian distance in Eq. (3) is symmetric at the pairwise point level. However, this does not make the two terms in Eq. (5) identical, because the asymmetry is introduced at the **batch-wise normalization level**, not by the distance function itself.
>
> Specifically, let $D_{ij} = d_{L,c}(\tilde S_i, \tilde T_j)$. Then $L_{HCL}(\tilde S, \tilde T)$ corresponds to skeleton-to-text contrastive learning, where each skeleton anchor is normalized over all text candidates in the batch, i.e., row-wise normalization of $D$. In contrast, $L_{HCL}(\tilde T, \tilde S)$ corresponds to text-to-skeleton contrastive learning, where each text anchor is normalized over all skeleton candidates, i.e., column-wise normalization of $D$.
>
> These two objectives are generally different even if $d_{L,c}(x,y) = d_{L,c}(y,x)$, because this symmetry does not imply $d_{L,c}(\tilde S_i,\tilde T_j) = d_{L,c}(\tilde S_j,\tilde T_i)$. In other words, symmetry of the pointwise distance does not imply symmetry of the batch-wise contrastive loss. We will revise the manuscript to clarify this point around Eq. (5), so that the intended bidirectional contrastive formulation is more explicit.
>
>
>
> **[Writing]** Thank you for this suggestion. We agree that the method section is currently too dense. In the revision, we will reorganize it with a clearer pipeline overview, explicitly separate the roles of EH-HCLIP and LMV-Cache, and distinguish training from novel-class inference more clearly. We will also keep only the most essential equations in the main text and move longer derivations and implementation details to the appendix, so that the overall pipeline becomes easier to follow.

---

> > ### Author Rebuttal · Reviewer_zRie · 2026-04-01
> >
> > I appreciate the authors addressing my concerns. All my questions are resolved.

---

> > > ### Author Response · Authors · 2026-04-03
> > >
> > > We sincerely thank the reviewer for the thoughtful comments and careful consideration of our work. We are grateful that our responses were helpful and that all concerns have been resolved.

---

### Official Review · Reviewer_DNfg · 2026-03-12

**Soundness:** 3
**Presentation:** 3
**Significance:** 4
**Originality:** 3
**Overall Recommendation:** 4
**Confidence:** 3

**Summary:**

The paper proposes SkelHCC for one-shot skeleton-based action recognition. It integrates two main components: an Explicitly Hierarchical Hyperbolic CLIP module (EH-HCLIP) that embeds skeleton and language modalities into a shared hyperbolic space, and a training-free LLM-guided Multi-granularity Voting Cache (LMV-Cache) that reuses the learned hyperbolic space for test-time adaptation. Results on three challenging datasets show improvements over existing SOTA methods.

**Compliance With Llm Reviewing Policy:**

Affirmed.

**Final Justification:**

The rebuttal has addressed my main concerns, and I will keep my rating.

**Key Questions For Authors:**

1. In Table 5, how are the learnable-mask and attention-mask baselines trained in the one-shot novel-class setting?
2. In Equation 4, why is the denominator summed over  j ≠ i rather than all j?

**Limitations:**

No. The framework depends on LLM-generated semantic masks at inference time, which depends on the LLM's prior knowledge of those actions. If the action label is ambiguous or outside the LLM's training distribution, the mask may be misleading.

**Strengths And Weaknesses:**

Strengths:
1. The paper is well-motivated. One-shot skeleton action recognition is challenging, and using language priors is a reasonable direction for improving generalization.
2. The overall system design is coherent. EH-HCLIP and LMV-Cache are complementary, and Figure 1 is clear.
3. The experimental results show significant improvements on three challenging benchmarks.

Weaknesses:
1. The LMV-Cache requires LLM-generated importance masks at inference time. No failure case analysis when there is an out-of-LLM-distribution action. Will the LLM prior be misleading for novel classes at test time?
2. The ablations do not cleanly separate geometry from hierarchy. Table 3 compares Euclidean CLIP, HCLIP, and EH-HCLIP, but there is no Euclidean version with the same explicit BJ/BP/FB hierarchy. The results cannot demonstrate that how much of the gain comes from hyperbolic geometry, and how much comes from simply adding a better hierarchical decomposition.

---

> ### Author Rebuttal · Authors · 2026-03-31
>
> We thank the Reviewer DNfg for the careful and constructive comments. We have carefully considered and addressed each concern point by point in the rebuttal.
>
> **[Case analysis]** We acknowledge that LLM priors may be imperfect for novel or ambiguous actions. However, our design mitigates this in two ways:
>
> - The masks are used only as soft weights, not hard constraints, so the model primarily relies on data-driven skeleton features.
> - The final prediction is based on multi-granularity fusion (FB/BP/BJ), which provides robustness even if some part-level priors are suboptimal.
>
> Empirical evidence further supports robustness:
>
> - **Ablations (Tab. 5, Tab. R1):** replacing LLM masks with random, uniform, or label-shuffled masks degrades performance (e.g., 71.6 → 68.6), indicating that correct priors help, while the model remains stable when they are imperfect.
>
> - **Controlled perturbations (Tab. R1):** suppressing high-weight regions causes a larger drop than suppressing low-weight ones (71.6 → 68.2 vs. 69.8), confirming that masks capture discriminative regions.
>
>   **Tab. R1. Sanity checks.** *(w/o mask: 68.5)*
>
>   | Method  |  BJ (k=5) |  BP (k=1) |BJ+BP |
>   | -| -| -| -|
>   | Correct Mask  |  69.6 |  69.9 | 71.6 |
>   | Label-shuffled Mask | 67.9 | 67.5 | 68.6 |
>   | Top-k Suppression | 68.3 ↓1.3 | 68.0 ↓1.9 | 68.2 ↓3.4 |
>   | Low-k Suppression | 69.3 ↓0.3 | 69.5 ↓0.4 | 69.8 ↓1.8 |
>
> - **Case studies and failure analysis (Tabs. R2–R3):**  Gains are largest on actions with clearer body-region or motion grounding from labels (e.g., *tear up paper*, *reach into pocket*), while failures  (e.g.,  *use a fan*) suggest that LLM priors are less reliable for subtle, object-dependent, or semantically broad actions. Notably, when BP-level priors are imperfect, adding BJ-level evidence can recover performance (Tab. R3), suggesting a coarse-to-fine compensation mechanism and robustness to noisy LLM priors.
>
> **Tab.R2. Gains with LLM masks/Hierarchy.**
>
> | Methods|  Tear up paper | Sniff (smell) | Reach into pocket|Hush|Open a box | Use a fan / feeling warm |
> |-|-|-|-|-|-|-|
> |LMV-Cache w/o BP,BJ Hierarchy|27.21|23.35|57.67|85.92|13.64|63.07
> |LMV-Cache w/o LLM Mask|30.84|25.03|60.74 | 87.39 |13.85|63.49|
> |LMV-Cache|36.92 (+6.08)|30.36 (+5.33)|64.86 (+4.12)|86.76 (-0.63)|12.59 (-1.26)|60.84 (-2.65)|
>
> **Tab. R3. Case-level evidence that hierarchy mitigates imperfect BP-level LLM priors.**
>
> | Variant (FB Cache) | reach into pocket | put on headphone|
> |-|-|-|
> |+ BP Hierarchy + BP Uniform mask | 58.10 | 85.87 |
> |+ BP Hierarchy + BP LLM mask |    56.30 (-1.80↓) | 83.62 (-2.25↓) |
> |+ BP, BJ Hierarchy + BP LLM mask + BJ Uniform mask | 59.79 (+3.49↑) |   88.01 (+4.39↑) |
> |+ BP, BJ Hierarchy + BP, BJ LLM masks (LMV-Cache)  | 64.87 (+5.08↑) |   89.29 (+1.28↑) |
>
>
>
> **[Geometry and Hierarchy Gains]** We thank the reviewer for this important comment. We add **EH-CLIP**, which uses the same explicit BJ/BP/FB hierarchy as EH-HCLIP but remains in Euclidean space. Table R4 shows that both geometry and hierarchy contribute: removing either one causes a clear drop, showing that hyperbolic geometry and explicit hierarchy are complementary.
>
> **Tab.R4. Disentangling geometry and hierarchy.**
>
> | Method                                | Configuration        | Acc.        |
> | -| ---- | ----------- |
> | Proposed SkelHCC                      | EH-HCLIP + LMV-Cache | 71.6        |
> | w/o Hierarchy in training and testing | HCLIP + Cache        | 64.8 (↓6.8) |
> | w/o Hierarchy in training             | HCLIP + LMV-Cache    | 68.2 (↓3.4) |
> | w/o Hyperbolic space                  | EH-CLIP + LMV-Cache  | 68.0 (↓3.6) |
>
>
>
> **[Mask Baseline]** We thank the reviewer for this important question. The Learnable Mask replaces the LLM-derived BJ/BP masks with trainable mask parameters, and the Attention Mask replaces them with a lightweight self-attention-based mask generator over support skeleton features. For fairness, both are trained only on the base-class dataset with cross-entropy, while the skeleton backbone and CLIP text encoder remain frozen. During novel-class one-shot evaluation, no mask parameters are updated and no query labels are used. Thus, they differ from our method only in how the masks are obtained, while the rest of the framework and protocol remain unchanged. We will clarify this in the manuscript.
>
>
>
> **[Eq.(4)]** Thank you for the careful observation. In Eq. (4), we incorrectly followed a notation used in some prior contrastive learning works, where terms such as $j \neq i$ are introduced to exclude trivial self-comparisons. However, after re-checking Eq. (4) and our implementation, we agree that the denominator should sum over all candidates in the batch, rather than only ($j \neq i$). In the implementation, we use the standard PyTorch *nn.functional.cross_entropy* loss, which normalizes over all candidates in the batch. Therefore, this is a typo in the manuscript. We thank the reviewer for pointing this out and will correct Eq.(4) in the revision.

---

> > ### Author Rebuttal · Reviewer_DNfg · 2026-04-03
> >
> > I thank the authors for the rebuttal. This time, I will maintain the score.

---

> > > ### Author Response · Authors · 2026-04-03
> > >
> > > We sincerely thank the reviewer for the thoughtful comments and careful consideration of our work. We are grateful that our responses were helpful and that all concerns have been resolved.

---

### Official Review · Reviewer_B6Lq · 2026-03-13

**Soundness:** 3
**Presentation:** 3
**Significance:** 3
**Originality:** 3
**Overall Recommendation:** 4
**Confidence:** 4

**Summary:**

This paper proposes SkelHCC, a framework for one-shot skeleton action recognition. It has two main parts: (i) an EH-HCLIP module that uses hierarchical skeleton information (full body, body parts, joints) for contrastive learning in hyperbolic space to align skeleton and text, and (ii) a training-free LMV-Cache module that uses LLM-generated action descriptions to create body part importance masks for better retrieval. Experiments on NTU-60, NTU-120, and PKU-MMD show the method outperforms prior approaches in one-shot settings.

**Compliance With Llm Reviewing Policy:**

Affirmed.

**Final Justification:**

The rebuttal addressed my concerns with additional experimental results, I raise my score and encourage the author to include these analyses and real-world eavaluations in the revision.

**Key Questions For Authors:**

Please see above Concerns.

**Limitations:**

The paper does not fully discuss the limitations. The per-class analysis on the specific activities that benefit the least/most from the proposed model could be include to help understand the limitations.

**Strengths And Weaknesses:**

Strengths:

+ The motivation is clear and the paper is easy to follow.

+ Using LLMs to generate body part masks is a effective way to inject prior knowledge without extra annotations, and the method is stable across different LLMs.

+ The experiments show clear gains over previous SoTA (CrossGLG) on the NTU-60, NTU-120 and PKU-MMD datasets.

Concerns:

- While this paper applies the skeleton decomposition idea to hyperbolic contrastive learning and cache models, However, the decomposition approach itself is not new from recent works [1*, 2*, 3*]. The author needs to explain more clearly how their decomposition method differs from or improves upon these existing works.

- The LLM mask generation process shows mask weights but does not explain how the LLM infers part importance from action names or evaluate mask quality.

- The paper only reports overall accuracy without showing which actions benefit from the method and which do not. For example, does the multi-granularity design actually help distinguish similar actions? Do the LLM-generated masks really guide the model to focus on relevant body parts? Without case studies or failure analysis, it is hard to tell if the claimed problems are actually solved.

- The datasets used are limited to NTU and PKU-MMD, which are relatively controlled laboratory environments. More complex and realistic datasets like Kinetics or UAV-Human would better test the effectiveness in real-world scenarios where existing methods struggle .

- The model complexity and inference speed are not discussed.

References：

[1*] Part-aware Unified Representation of Language and Skeleton for Zero-shot Action Recognition. CVPR, 2024.

[2*] Fine-Grained Side Information Guided Dual-Prompts for Zero-Shot Skeleton Action Recognition. ACM MM, 2024.

[3*] Joint-Partition Group Attention for skeleton-based action recognition. Signal Processing, 2024.

---

> ### Author Rebuttal · Authors · 2026-03-31
>
> **[Decomposition.]** We sincerely thank you for the valuable comments. While skeleton decomposition is not new [1–3], our contribution is not the decomposition. Instead, it’s SkelHCC, a new framework with two modules that explicitly and jointly leverages a shared hierarchy for OSAR.
>
> Unlike prior work that uses decomposition mainly for representation learning, we reuse a single hierarchy across both training and inference: EH-HCLIP enables hyperbolic cross-modal alignment, and LMV-Cache performs LLM-guided, training-free retrieval.
>
> Specifically, the difference includes:
>
> + **Hierarchy.** We define an explicit joint→part→whole structure (BJ/BP/FB), unlike prior implicit or task-specific designs (e.g., global/local [1], topology-based parts [2], adaptive grouping [3]). This shared hierarchy induces more structured geometry (Tab.11).
>
> + **Usage.** In prior work, decomposition is mainly limited to training-time representation learning. Our hierarchy plays a dual role, bridging training (alignment) and inference (retrieval), which is key for OSAR.
>
> **[Mask Inference and Quality]**  We thank the reviewer for this important question and clarify how masks are inferred and validated.
>
> **(1) Inference:** The LLM maps action labels to body-part importance (in Fig. 2 and Fig. 6) using the BJ/BP/FB hierarchy (e.g., kicking → legs) . Masks are used as semantic priors, not ground-truth saliency, and reweigh similarities in LMV-Cache (Eq. 9).
>
> **(2) Quality:** We evaluate masks through their effect on retrieval.  Masks improve accuracy (68.5→71.6), while random/shuffled masks degrade it (Tab.5, R1). Gains are consistent across LLMs and runs (Tab. 10, Fig. 7), and visualizations align with action-relevant regions (Fig. 2).
>
> **(3) Sanity checks (Tab.R1):**
>
> + Label-shuffled masks reduce performance (71.6 → 68.6)
> + Removing high-weight regions causes larger drops than low-weight ones (71.6 → 68.2 vs. 69.8), confirming discriminative validity.
>
> **Table R1. Sanity checks.** *(w/o mask: 68.5)*
> | Method | BJ (k=5) |  BP (k=1) | BJ+BP|
> |-|-|-|-|
> | Correct Mask | 69.6 | 69.9 | 71.6 |
> | Label-shuffled Mask | 67.9 | 67.5 | 68.6 |
> | Top-k Suppression |68.3 ↓1.3 | 68.0 ↓1.9 | 68.2 ↓3.4 |
> | Low-k Suppression |69.3↓0.3 | 69.5 ↓0.4 | 69.8 ↓1.8 |
>
> **[LLM Guidance]**
>
>  **(1) Multi-granularity design.** Tab. R2 shows that BP/BJ hierarchy improves fine-grained actions that are hard to distinguish from global appearance alone, e.g., *tear up paper* (27.21→30.84→36.92) and *reach into pocket* (57.67→60.74→64.86), confirming the benefit of localized body-part/joint cues.
>
> **(2) Relevant body-part guidance.** Fig. 2 and Tab. 9 show that the predicted masks/part responses align with action-relevant body regions. Random or shuffled masks degrade performance (Tab. 5, R1), indicating that the LLM masks provide meaningful rather than arbitrary guidance.
>
> **(3) Failure analysis.** Table R3 shows cases such as *reach into pocket* and *put on headphone*, where coarse BP priors are insufficient. Adding BJ evidence improves prediction, highlighting the value of coarse-to-fine fusion.
>
> **Table R2. Top- and low-gain novel classes with LLM masks.**
>
> | Methods|  Tear up paper | Sniff (smell) | Reach into pocket|Hush (quiet) |    Open a box | Use a fan / feeling warm |
> |-|-|-|-|-|-|-|
> |LMV-Cache w/o BP,BJ Hierarchy|27.21|23.35|57.67|85.92|13.64|63.07
> |LMV-Cache w/o LLM Mask|30.84|25.03|60.74 | 87.39 |13.85|63.49|
> |LMV-Cache|36.92 (+6.08)|30.36 (+5.33)|64.86 (+4.12)|86.76 (-0.63)|12.59 (-1.26)|60.84 (-2.65)|
>
> **Table R3. Case-level evidence that hierarchy mitigates imperfect BP-level LLM priors.**
>
> | Variant (FB Cache) |reach into pocket| put on headphone |
> |-|-|-|
> | + BP Hierarchy + BP Uniform mask | 58.10 | 85.87 |
> | + BP Hierarchy + BP LLM mask | 56.30 (-1.80↓) | 83.62 (-2.25↓) |
> | + BP, BJ Hierarchy + BP LLM mask + BJ Uniform mask | 59.79 (+3.49↑) | 88.01 (+4.39↑) |
> | + BP, BJ Hierarchy + BP, BJ LLM masks (LMV-Cache)  | 64.87 (+5.08↑) | 89.29 (+1.28↑) |
>
>  **[Dataset]** Table R4 shows the results on the Kinetics dataset.
>
> **Table R4.  Kinetics**
>
> | Method |20-base|40-base|
> |-|-|-|
> |MotionBERT|13.2|16.6|
> |InfoGCN|13.3|18.2|
> |CrossGLG|17.4|19.2|
> |SkelHCC (ours)|21.9|23.1|
>
> **[Overhead]**
>
> **Efficiency.** Minimal cost: +0.318 MFLOPs (~0.008%) over 3.87 GFLOPs, with real-time speed (169.5 FPS).
>
> **Trade-off.** Significant gains (67.6→71.6) with negligible overhead.
>
> **Practicality.** Training-free inference, offline LLM masks, <1GB memory, and only tens of ms overhead.
>
> **Table R5. Inference cost.**
> |Variant|Acc (%) | FLOPs / query |FPS|
> |-|-|-|-|
> | EH-HCLIP| 59.1|3.87 G| 202.59 |
> | + Cache (FB)|67.6|+0.020 M| 187.26 |
> | + Cache (FB + BP)|68.9|+0.062 M|179.36|
> | + Cache (FB + BJ)|68.4|+0.277 M|181.43|
> | + LMV-Cache (FB + BJ + BP) | 71.6 (+12.5) |+0.318 M | 169.49 |
>
> **Table R6. Adaptation  memory.**
> | Process| Peak GPU Mem (GB) | Time / episode (ms)|
> |-|-|-|
> | Whole adaptation pipeline|0.95|379.66 |
> | Cache construction only (LMV-Cache) |0.79 | 77.40|

---

> > ### Author Rebuttal · Reviewer_B6Lq · 2026-04-04
> >
> > The rebuttal addressed my main concerns with additional experiments and analysis. I will therefore consider adjusting my score accordingly.

---

> > > ### Author Response · Authors · 2026-04-04
> > >
> > > We thank the reviewer for the thoughtful comments and careful consideration of our work. We are glad that the additional experiments and analysis have addressed the concerns, and we appreciate your consideration in revising the score.

---

### Decision · Program_Chairs · 2026-04-30

**Decision:**

Accept (regular)

**Comment:**

This paper received three weak accept and one weak reject as the final rating. Overall, reviewers feel the rebuttal has helped to address most previous concerns. The negative reviewer also mentioned "The authors have addressed most of my concerns." For the remained concerns of computational overhead, authors provide extra inputs. Overall this is a solid paper. AC follows majority of reviewers to accept.